# Control of RNA viruses in mosquito cells through the acquisition of vDNA and endogenous viral elements

Michel Tassetto[1†], Mark Kunitomi[1†‡], Zachary J Whitfield[1], Patrick T Dolan[1], Irma Sánchez-Vargas[2], Miguel Garcia-Knight[1], Isabel Ribiero[1], Taotao Chen[1], Ken E Olson[2]*, Raul Andino[1]*

[1]Department of Microbiology and Immunology, University of California, San Francisco, San Francisco, United States; [2]Department of Microbiology, Immunology and Pathology, Arthropod-borne and Infectious Diseases Laboratory, Colorado State University, Fort Collins, United States

**Abstract** *Aedes aegypti* transmit pathogenic arboviruses while the mosquito itself tolerates the infection. We examine a piRNA-based immunity that relies on the acquisition of viral derived cDNA (vDNA) and how this pathway discriminates between self and non-self. The piRNAs derived from these vDNAs are essential for virus control and Piwi4 has a central role in the pathway. Piwi4 binds preferentially to virus-derived piRNAs but not to transposon-targeting piRNAs. Analysis of episomal vDNA from infected cells reveals that vDNA molecules are acquired through a discriminatory process of reverse-transcription and recombination directed by endogenous retrotransposons. Using a high-resolution *Ae. aegypti* genomic sequence, we found that vDNAs integrated in the host genome as endogenous viral elements (EVEs), produce antisense piRNAs that are preferentially loaded onto Piwi4. Importantly, EVE-derived piRNAs are specifically loaded onto Piwi4 to inhibit virus replication. Thus, *Ae. aegypti* employs a sophisticated antiviral mechanism that promotes viral persistence and generates long-lasting adaptive immunity.
DOI: https://doi.org/10.7554/eLife.41244.001

*For correspondence:
Kenneth.Olson@colostate.edu
(KEO);
raul.andino@ucsf.edu (RA)

†These authors contributed
equally to this work

Present address: ‡IBM Almaden
Research Center, San Jose,
United States

Competing interests: The
authors declare that no
competing interests exist.

Reviewing editor: Neal
Silverman, University of
Massachusetts Medical School,
United States

## Introduction

*Aedes aegypti* are vectors of some of the world's most widespread and medically concerning pathogens such as yellow fever, dengue, Zika and chikungunya viruses. Although arboviruses can cause severe disease in humans, they generally cause non-cytopathic, persistent, lifelong infections of competent *Aedes* spp. vectors. Antiviral immunity is thought to be central to these divergent outcomes. As in other arthropods, RNA interference (RNAi) is a major component of the mosquito antiviral defense (*Blair and Olson, 2015*). Intracellular long dsRNA, such as viral replicative intermediates generated during positive-sense viral RNA replication, are processed by the processive endoribonuclease Dicer2 (Dcr2) into 21 nucleotide (nt) small interfering RNA (siRNAs) duplexes, which are loaded onto the endoribonuclease Argonaute-2 (Ago2) at the core of the RNA-induced silencing complex (RISC). Ago2 then cleaves one strand of the virus-derived siRNA duplex and utilizes the remaining strand as a guide to target and cleave complementary viral RNAs, thereby restricting viral replication (*van Rij et al., 2006*).

In addition to 21 nt viral siRNAs (v-siRNAs), arboviral infections of *Aedes* spp. mosquitoes and cultured cells also lead to the accumulation of 24–30 nt long virus-derived small RNAs (*Scott et al., 2010*; *Hess et al., 2011*; *Morazzani et al., 2012*; *Vodovar et al., 2012*; *Goic et al., 2016*; *Miesen et al., 2015*). These small RNAs are similar in size to PIWI-interacting small RNAs (piRNAs), which are generally associated with germline defense against mobile genetic elements. Like

germline piRNAs, virus-derived piRNA (v-piRNA) production involves PIWI proteins (*Miesen et al., 2015*; *Varjak et al., 2017a*). In the germline, long antisense piRNA transcripts from genomic piRNA clusters are cleaved to produce primary guide piRNAs with a uridine at their 5' end (guide U1), which are loaded onto a Piwi protein (Piwi or Aubergine in *Drosophila* [*Brennecke et al., 2007*]) to target and cut cognate transposon mRNAs. Despite the absence of pairing between the U1 of piRNAs bound to Piwi/Aubergine and the target mRNA, Piwi/Aub preferential binds to target sequences with an adenine at their first position (target A1) (*Wang et al., 2014*). The 3' products from cleaved transposon mRNAs are used to form secondary guide piRNAs with an adenine at their new position 10 (secondary guide A10) and are loaded onto Ago3. These secondary piRNA complexes can then target their corresponding antisense transcript to generate a new antisense U1 piRNA, which are loaded onto another Piwi/Aub protein. This ping-pong model thereby provides the germ line with a pathway for biogenesis of anti-transposon piRNAs. In arbovirus-infected cells, the v-piRNAs also present characteristics supporting the idea that these viral small RNAs are produced via a ping-pong mechanism (*Vodovar et al., 2012*). However, v-piRNAs are not observed in virus-infected Drosophila, nor are PIWI proteins involved in the fly antiviral defense (*Petit et al., 2016*). Despite a much greater invasion of its genome by TEs compared to the fruitfly, the *Aedes* mosquito dedicates a smaller fraction of its piRNAs to target TEs (*Arensburger et al., 2011*) and possess an expanded Piwi gene family compared to *Drosophila* (*Lewis et al., 2016*), suggesting a functional diversification of the Piwi family in mosquito. Conflicting reports have attributed antiviral roles to different PIWI clade proteins in mosquito. For instance, Piwi5 and Ago3 have been suggested to be specifically required for the formation of virus-derived piRNAs, although knock-down of these proteins had no effect on viral replication (*Miesen et al., 2015*; *Varjak et al., 2018*). By contrast Piwi4, has been shown to be antiviral in dsRNA knock-down screens (*Varjak et al., 2018*; *Varjak et al., 2017a*) but a direct association of viral piRNAs with Piwi4 has not been shown yet (*Miesen et al., 2015*).

Upon arboviral infection in insect cells, fragments of viral RNA genomes are converted into vDNA by the reverse-transcriptases derived from endogenous retrotransposons, leading to the accumulation of episomal vDNA molecules. These episomal vDNA molecules serve in the amplification of the antiviral RNAi response in *Drosophila* (*Tassetto et al., 2017*) and have been linked to vector competence in mosquito (*Goic et al., 2016*). At the genomic level, integration of viral elements has been identified in all eukaryotes and for all virus families (*Holmes, 2011*). In insects, these endogenous viral elements (EVEs) are derived mostly from non-retroviral RNA viruses (*Palatini et al., 2017*; *Whitfield et al., 2017*), with *Aedes spp* genomes harboring approximately 10 times more EVEs than other mosquito species. Parallel analyses of genome and small RNA sequences from *Aedes* mosquitoes and derived cell lines reveal that EVEs organize in large loci characterized by high LTR retrotransposon density and the production of high levels of piRNAs (*Whitfield et al., 2017*). The abundance of EVEs and EVE-derived piRNAs in mosquito cells, the accumulation v-piRNAs during infection and the contribution of PIWI proteins in antiviral defense suggest the existence of EVE-derived immunity in mosquito through RNAi mechanisms. However, from a mechanistic point of view, little is known about the role of the piRNA machinery in the RNAi-based adaptive immunity in mosquito and the potential antiviral function of EVEs.

Here, we demonstrate that Piwi4 is upregulated in somatic tissues in adult female *Ae. aegypti* following blood meal and restricts dengue virus replication in vivo. In infected cells, Piwi4 is required for the accumulation of mature v-piRNAs and binds preferentially to antisense v-piRNAs (corresponding to the anti-genome viral RNA) and not to anti-transposon piRNAs. High-throughput sequencing of episomal DNA accumulating during infection reveals that viral RNAs are preferentially reverse-transcribed, indicating that this antiviral system discriminates between self (host mRNAs) and non-self RNAs (viral RNA) during the vDNA acquisition process. Furthermore, Piwi4 preferentially binds to v-piRNAs derived from genomic EVEs and as compare to v-piRNAs derived from replicating viral RNA. Consistent with this observation, knock-down of Piwi4 results in a decrease in EVE-derived piRNAs and an increase of CFAV RNA accumulation. Furthermore, Aag2 cells infected with Sindbis virus engineered to carry EVE sequences lead to efficient inhibition of SINV replication in a sequence and strand-specific manner, and knock-down of Piwi4 upregulates virus replication. Together our results indicate that genomic EVEs, EVE-derived piRNAs and Piwi4 are central to the antiviral adaptive immunity in *Ae. aegypti* mosquitoes. Given that EVEs are stably incorporated into genome, these observations suggest that the acquisition of vDNA provide a mechanism to long-lasting mosquito adaptive immunity.

## Results

### Piwi4 is required to restrict virus replication in mosquitoes

In *Ae. aegypti*, four Piwi clade proteins have been associated with the host response to arboviral infection. While Piwi5 and Ago3 are involved in v-piRNA production, their knock down does not affect viral replication (*Schnettler et al., 2013*; *Miesen et al., 2015*). By contrast, knock-down of Piwi4 expression leads to increased replication of Semliki Forest Virus, Bunyawera virus, Zika virus and Rift Valley fever virus but does not affect the overall amount of v-piRNAs (*Varjak et al., 2017a*; *Dietrich et al., 2017*; *Varjak et al., 2017b*). We first confirmed the role of PIWI proteins in antiviral defense using a RNAi screen against all seven Aedes Piwi transcripts, followed by Sindbis virus infection (SINV; *Alphavirus; Togaviridae*). We found that Piwi4 had the strongest antiviral effect in Aag2 cells (*Figure 1—figure supplement 1A*). Likewise, Piwi4 knock-down in Aag2 cells led to higher viral replication of dengue virus type 2 (DENV2; *Flavivirus; Flaviviridae*) and Chikungunya virus (CHIKV; *Alphavirus*) (*Figure 1A*, n = 4, Mann-Whitney U test, p<0.05).

All previous studies of Piwi proteins in mosquito antiviral defense have been conducted in cell culture. To relate these findings to the actual insect vector, we examined Piwi4 expression in somatic tissues of adult female *Ae. aegypti* mosquitoes and observed transient increases in both midguts and carcasses, but not in the ovaries, after a blood meal (*Figure 1b*, n = 4, Mann-Whitney U test, p<0.05). The antiviral role of Piwi4 in female *Ae. aegypti* was then tested by anti-Piwi4 dsRNA injection (dsPiwi4) followed by infection with DENV-2 (Jamaica 1409) by blood meal. Depletion of Piwi4 (dsPiwi4, *Figure 1—figure supplement 1B*, Mann-Whitney U test, n = 4, p≤0.01–0.05, double and single asterisks respectively) resulted in a significant increase of DENV2 genomic RNA in both the midgut and carcass at 7 and 10 days post infection (dpi) and a significant increase of infectious virus titers in whole mosquitoes at 10 dpi (*Figure 1c*, n = 4, Mann-Whitney U test, p≤0.05). These results indicate that Piwi4 mediates antiviral defense not only in cultured Aag2 cells but also in *Ae. aegypti* somatic tissues.

### Piwi4 is required for v-piRNA maturation

We next examined the specific role of Piwi4 in antiviral defense. To determine whether Piwi4 is involved in the production of mature antiviral small RNAs, we analyzed virus-derived small RNAs in Aag2 cells infected with SINV, in which Ago2, Piwi4 or Ago3 were depleted using specific dsRNAs (*Figure 2—figure supplement 1A*). In control dsRNA-treated cells, both SINV-derived siRNAs (v-siRNAs, 21 nt long) and SINV-derived piRNAs (v-piRNAs, 24–30 nt long) accumulated during infection (*Figure 2B*). The 21 nt v-siRNAs, are uniformly derived from the entire viral genome and correspond to both the sense and antisense strands of the genome (*Figure 2—figure supplement 1B*). In contrast, 24 to 30 nt v-piRNAs were asymmetrically distributed with respect to strand polarity and position on the virus genome (*Figure 2—figure supplement 1C*). As previously described (*Miesen et al., 2015*; *Varjak et al., 2017a*), the overall size distribution of v-piRNAs was not affected by Piwi4 knock down. We then asked whether v-piRNA maturation was affected by Piwi4 loss-of-function. Maturation of piRNAs is mediated by their 3'-end methylation following loading onto a PIWI protein and can be readily assessed by its resistance to ß-elimination (*Horwich et al., 2007*). Analysis of virus-derived small RNAs after ß-elimination showed a decrease in mature v-piRNAs in Piwi4 knock-down samples (*Figure 2B*), suggesting a role for Piwi4 in v-piRNA maturation.

The large accumulation of sense v-piRNAs from the 5'-end of SINV subgenomic RNA (genome 7611–11703 nt) is consistent with a model in which piRNA ping-pong amplification correlates with the abundance of RNA target (for instance, SINV subgenomic RNA) (*Figure 2—figure supplement 1C*). To accurately assess the effect of gene-knockdown on v-siRNAs and v-piRNAs maturation and avoid biases due to differences in viral RNA abundance (*Figure 2—figure supplement 1D*), read counts were normalized to SINV genome copy numbers (*Figure 2C*). Knockdown of either Piwi4 or Ago3 reduced the abundance of mature v-piRNAs (resistant to ß-elimination, *Figure 2D*) and transposon-derived piRNAs (*Figure 2—figure supplement 1E and F*). However, knockdown of Piwi4 or Ago2, but not Ago3, resulted in complete depletion of mature v-siRNAs (*Figure 2D*). Our results indicate that Piwi4 is involved in v-piRNA and v-siRNA maturation but appears to make a minor contribution to their initial production (see Discussion).

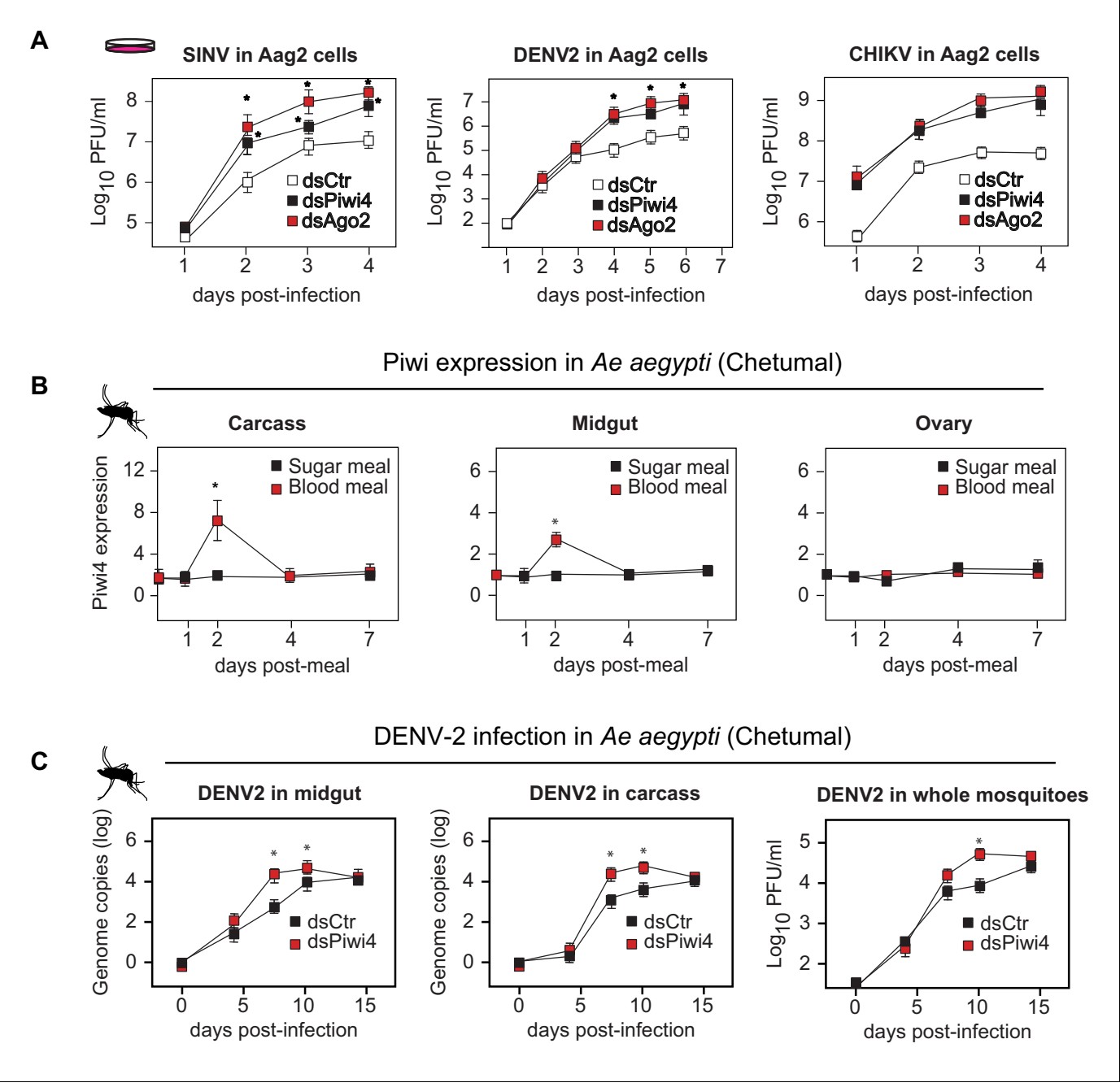

**Figure 1.** Piwi4 is a restriction factor in vivo, upregulated after blood meal. (A) Multi step growth curves of SINV, DENV-2 and CHIKV in Aag2 cells (MOI 0.1) treated with dsRNA targeting Ago2 or Piwi4 measured by plaque assay. Error bars depict standard deviation of four replicates. Significant changes over controls are marked with asterisks (p<0.05, Mann-Whitney U test). (B) Piwi4 mRNA measured by RT-qPCR from pools of either blood- or sugar- fed *Ae. aegypti* mosquitoes. The mean and standard deviation of four biological replicates of pools of five mosquitoes in carcass, midgut and ovary are shown. Significant changes over controls are marked with asterisks (p<0.05, Mann-Whitney U test). (C) Replication of DENV2 as measured by RT-qPCR or virus plaque assays. Mosquitoes were infected by intrathoracic injection with either dsPiwi4 or control dsRNA prior to infection. Error bars correspond to standard error of 20 (RT-qPCR) or 30 (plaque assays) biological replicates of individual mosquitoes. Significant changes over controls are marked with asterisks (p<0.05, Mann-Whitney U test).

DOI: https://doi.org/10.7554/eLife.41244.002

The following figure supplement is available for figure 1:

**Figure supplement 1.** Piwi4 knockdown in cell culture and adult female *Ae. aegypti* mosquitoes.

DOI: https://doi.org/10.7554/eLife.41244.003

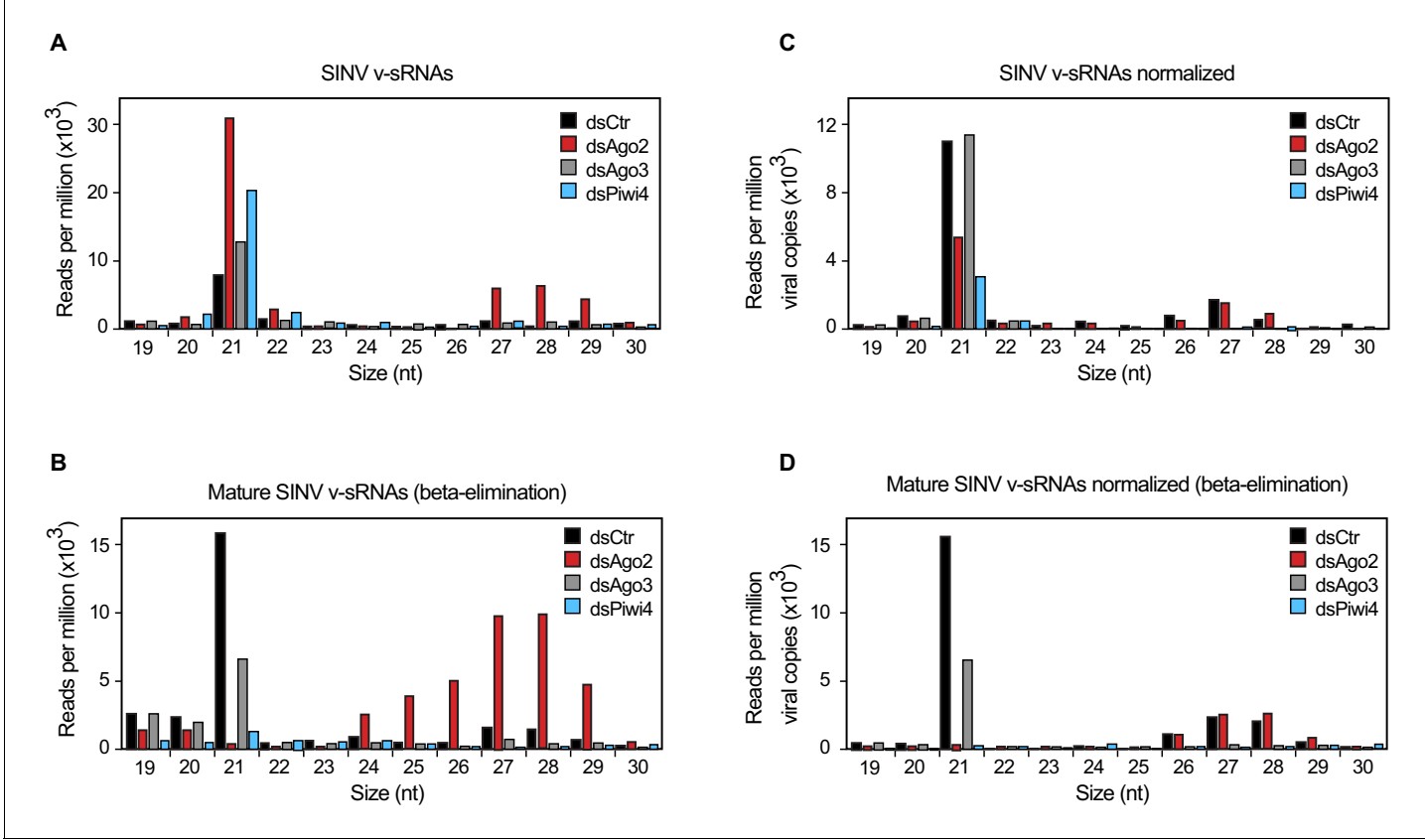

**Figure 2.** Piwi4 is required for v-piRNA maturation. (A) Size distribution plot of small RNAs mapping to the SINV genome from infected Aag2 cells treated with Fluc/control, Piwi4, or Ago3 dsRNA. (B) Size distribution plot of beta-elimination resistant small RNA mapping to the SINV genome from infected Aag2 cells treated with Fluc/control, Piwi4, or Ago3 dsRNA. (C) Same as in (A) but read counts were normalized to viral copy numbers. (D) Same as in (B) but read counts for beta-elimination resistant small RNAs were normalized to viral copy numbers.

DOI: https://doi.org/10.7554/eLife.41244.004

The following source data and figure supplement are available for figure 2:

**Source data 1.** Identification of small RNAs in the *Aedes aegypti* cell line Aag2 knocked-down for Ago2, Ago2 or Piwi4 expression and infected with Sindbis virus.

DOI: https://doi.org/10.7554/eLife.41244.007

**Figure supplement 1.** Effects of Ago2, Ago3 and Piwi4 knockdown on SINV-derived small RNAs.

DOI: https://doi.org/10.7554/eLife.41244.006

## Piwi4 binds to antisense v-piRNAs

During piRNA biogenesis, PIWI proteins associate with these small RNAs and facilitate their maturation (*Horwich et al., 2007*). To gain insights on th spectrum of piRNAs interacting with Piwi4 we ectopically expressed Piwi4-FLAG in Aag2 cells, infected with SINV and 48 hr post-infection, we analyzed cytoplasmic extracts by FLAG-immunoprecipitation (IP) followed by small RNA sequencing. To control for background resulting from the high abundance of small RNAs in the cells and potential confounding effects due to Piwi protein overexpression, we normalized small RNAs associated with Piwi4 to small RNAs present in the input sample. Contrary to previous reports that did not normalize to input small RNA content, we found that Piwi4 bound to v-piRNAs and that Piwi4-associated small RNAs were significantly enriched for antisense SINV-derived piRNAs (*Figure 3A*, antisense vs sense v-piRNAs, $^2$, p<2.2e-16). The enrichment of antisense v-piRNAs in Piwi4 pull-down (27–29 nt) was three fold higher than for other sizes of virus-derived small RNAs, including v-siRNAs (21-22nt). We assessed the variability within both input and pull-down samples by bootstrapping, where all SINV mapped reads in both datasets were resampled 10e4 times. Based on the bootstrap distribution, we calculated the 95% confidence intervals for the enrichment of each size of antisense small RNAs

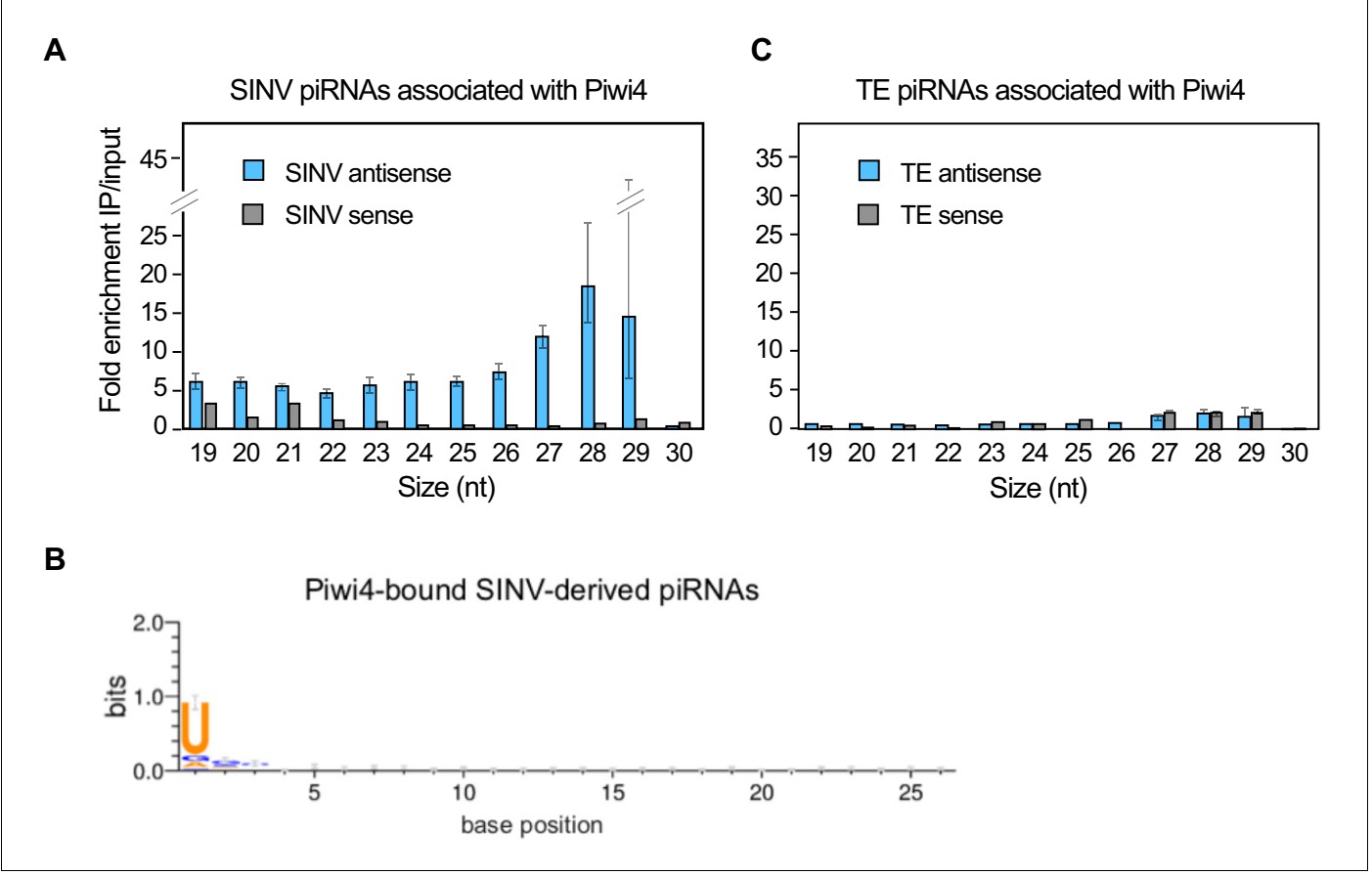

**Figure 3.** Piwi4 binds specifically to antisense v-piRNAs. (A) Enrichment of SINV-derived small RNA in Piwi4-immunoprecipitation (IP) fraction compared to input sample. (B) The base bias for each position of SINV-derived piRNAs co-immunoprecipitated with Piwi4-FLAG shows a Uracil bias at position 1, characteristic of antisense Piwi-associated piRNAs (shown in bits). (C) Same as in A but for transposons (TE)-derived small RNAs.

DOI: https://doi.org/10.7554/eLife.41244.008

The following source data and figure supplement are available for figure 3:

**Source data 1.** Identification of small RNAs preferentially bound by Piwi4 following Sindbis virus infection in the *Aedes aegypti* cell line Aag2.
DOI: https://doi.org/10.7554/eLife.41244.010

**Figure supplement 1.** Piwi4 binds to bona-fide v-piRNAs and interacts with known siRNA and piRNA pathway components.
DOI: https://doi.org/10.7554/eLife.41244.009

(error bars in *Figure 3A*) and found that they did not overlap between 27 and 28 nt antisense v-piRNAs and the all other antisense viRNAs. We thus conclude that antisense v-piRNAs were significantly enriched in Piwi4-bound fractions (at a 95% confidence level). In addition, Piwi4-associated v-piRNAs displayed a strong sequence bias, in which uracil is the most frequent base observed at the most-5' position (*Figure 3B*), a signature of Piwi-bound piRNAs. In contrast, Piwi4 did not significantly associate with TE-derived piRNAs (*Figure 3C*).

To further validate that Piwi4 preferentially bound to antisense v-piRNAs, we performed three additional IPs followed by small RNA sequencing experiments in SINV infected Aag2 cells: two using the Piwi4-FLAG construct and one negative control IP experiment in Aag2 cells transiently expressing eGFP. Deep-sequencing analysis of small RNAs in Piwi4-IP and input fractions confirmed that Piwi4 preferentially binds to antisense v-piRNAs (antisense vs sense v-piRNAs, $\chi^2$, p<2.2 e-16 and p<4.2e-9, for Experiment 1 and 2, respectively) with a U1 bias, during acute SINV infection in Aag2 cells (*Figure 3—figure supplement 1A*). In addition, v-piRNAs enrichment in Piwi4 IP experiments were significantly greater than in the eGFP control experiment (*Figure 3—figure supplement 1B*, $\chi^2$, p<1.497e-12 and p<2.2e-16 for 26–29 nt-long antisense v-piRNA enrichment in Exp1 vs

eGFP and Exp2 vs eGFP, respectively). Of note, normalization to input small RNA content revealed a lack of v-siRNA enrichment in the Piwi4-bound fraction, consistent with the size preference of PIWI proteins (*Brennecke et al., 2007*). This suggests that the large amount of v-siRNAs accumulating during SINV infection (*Figure 2A* and *Miesen et al., 2015*) was non-specifically immunoprecipitated with Piwi4.

These results further substantiate the idea that Piwi4 is functionally distinct from other Piwi proteins involved in defense against TE and viruses. In agreement with a recent study (*Varjak et al., 2017a*), we found that Piwi4 interacts with Ago2 (*Figure 3—figure supplement 1C*). Because knockdown of Piwi4 expression affects v-siRNAs production (*Figure 2D*) but does not specifically bind to them (*Figure 3—figure supplement 1A*), our results suggest that the siRNA and piRNA pathways are linked through Piwi4, which may act as a hub to coordinate production of antiviral small RNAs.

## The conversion of the viral RNA genome into vDNA in *Ae. aegypti* cells efficiently discriminates between viral and host RNAs

During infection of *Drosophila melanogaster* (*Tassetto et al., 2017*) and in mosquitoes (*Goic et al., 2016*; *Goic et al., 2013*; *Nag et al., 2016*), fragments of RNA virus genomes are reverse-transcribed early during infection, which results in viral cDNA formation (vDNA). We also observed that reverse transcriptase (RT) inhibitors AZT and Stavudine/d4T prevent production of vDNA in SINV-infected Aag2 cells (*Figure 4*). Furthermore, treatment with RT inhibitors resulted in an increase SINV replication (*Figure 4A*). Importantly, AZT and d4T treatments resulted in a significant reduction of v-piRNA reads but caused increased production of v-siRNAs (*Figure 4B*). These results provide a direct evidence that reverse transcription of viral genomes and production of piRNAs are essential components of the mosquito antiviral response. Indeed, even if siRNAs are normally produced in Piwi4-depleted cells, we observed a significant increase in SINV genomic RNA accumulation, thus Piwi4 and vDNA synthesis are essential for control of virus replication in mosquito cells. Our data, together with previous observations (*Goic et al., 2013*; *Goic et al., 2016*), suggest a model in which virus infection of mosquito cells induces the synthesis of vDNA, which in turn serves as a template for v-piRNA precursor production, which are then processed into mature antiviral v-piRNA by Piwi4 to repress viral replication (*Figure 4C*).

In plants and animals, retrotransposons produce circular DNA intermediates as part of their replication cycle (*Hotta and Bassel, 1965*; *Jones and Potter, 1985*; *Flavell and Ish-Horowicz, 1981*. We, and others (*Goic et al., 2016*; *Nag et al., 2016*), have hypothesized that vDNA in insects cells is produced in the context of retrotransposon replication (*Figure 4D*). In support of this model, we observed that PCR amplification of circular episomal DNA prepared from SINV-infected Aag2 cells with SINV genome-specific primers yielded a greater amplicon concentration than SINV-specific amplification of genomic DNA prep (*Figure 5—figure supplement 1A*). To directly analyze the role of retrotransposons in vDNA formation, we infected Aag2 cells with SINV and purified cytoplasmic extra-chromosomal DNA enriched for circular DNA (*Møller et al., 2016*), followed by Nextera cloning and Illumina deep-sequencing (*Figure 5A*). In addition to sequences derived from the circular mitochondrial genome (1% of total reads), we found that our preparations contained a significant enrichment of reads mapping to transposons (10–22% of total reads). Transposon sequences came almost exclusively from retrotransposons, as expected due to a cytoplasmic step in their replication cycle. LTR-retrotransposons (Ty1, Ty3 and Pao Bel) were the most abundant sub-class observed (*Figure 5B*). Importantly, we also observed more than 1200 reads in the circular DNA enriched fraction mapping throughout most of the SINV genome (*Figure 5D,i*) with two regions less covered by vDNA (between 3500–4000 nt and 7500–8000 nt), corresponding to the NSP2 ORF and the start of the subgenomic RNA.

Given that Aag2 cells are persistently infected with the insect-specific flavivirus, cell fusing agent virus (CFAV)(*Stollar and Thomas, 1975*), we reasoned that the circular DNA preparation should also contain vDNA derived from the CFAV genome. Indeed, we observed a large number of reads mapping throughout the CFAV genome (*Figure 5D,ii*).

Recombination of a retrotransposon transcript with a foreign RNA molecule during the cytoplasmic stage of replication can occur during reverse-transcription by a copy-choice mechanism (*Lai, 1992*; *Skalka, 2014*; *Zhang et al., 2000*). To determine whether recombination occurred between the RNA of an acutely infecting arbovirus and a mosquito retrotransposon transcript, we analyzed the sequencing read pairs of circular DNAs to identify hybrids between virus and

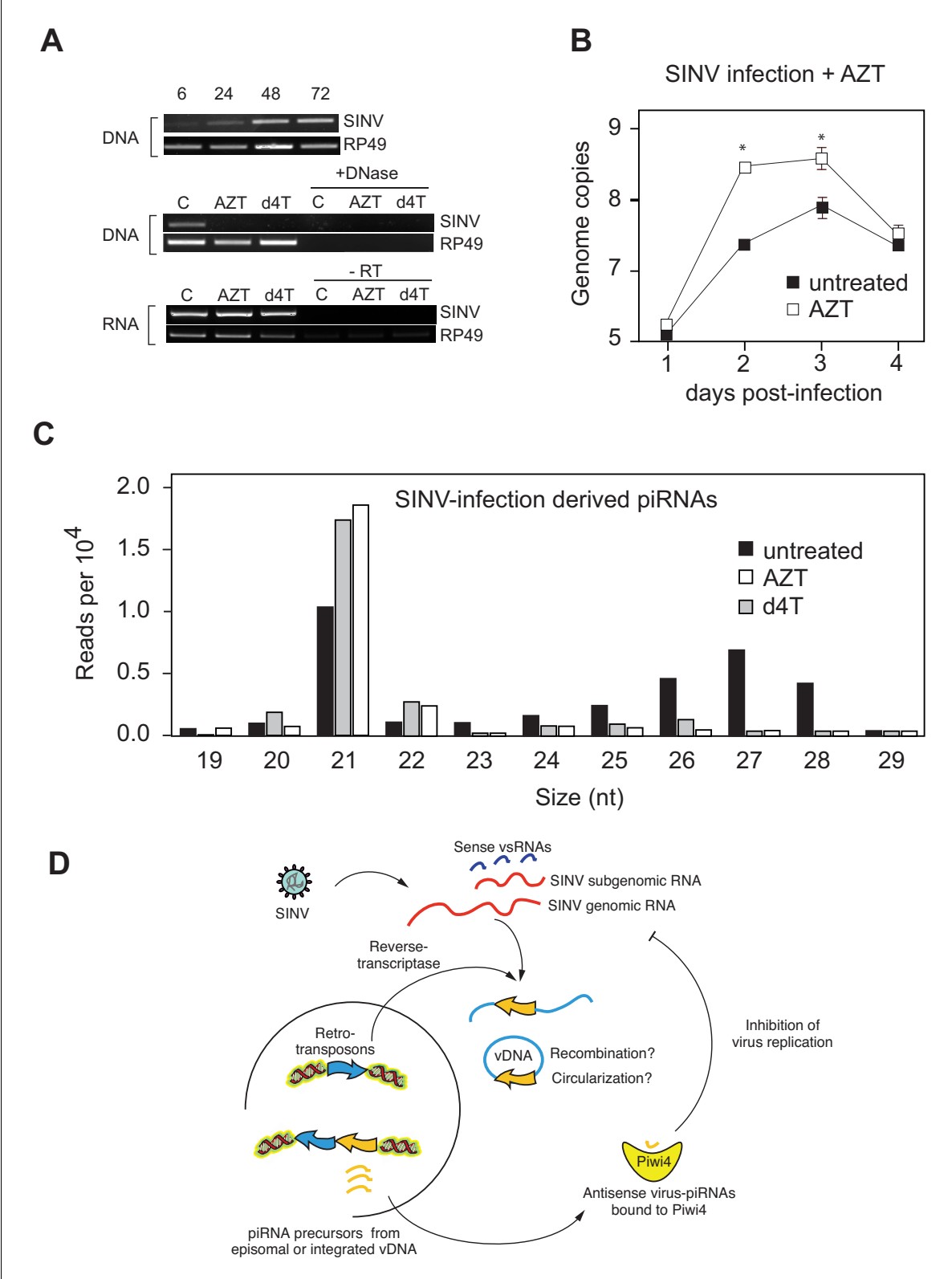

**Figure 4.** Production of v-piRNAs is blocked by reverse-transcriptase inhibitors. (A) Multi step growth curve of SINV RNA in Aag2 cells (MOI 0.1) with or without AZT treatment measured by RT-qPCR. RP49 was used as a normalization control. The error bars represent standard deviation of four biological replicates. Significant changes over untreated control are marked with asterisks (p<0.05, Mann-Whitney U test). B Size distribution plot of small RNAs mapping to the SINV genome from untreated infected Aag2 cells or cells treated with the RT-inhibitors AZT or d4T. Read counts per hundred were

*Figure 4 continued on next page*

*Figure 4 continued*

normalized to SINV genome copy numbers. (C) Schematic representation of the v-piRNA production. vDNA localized in the nucleus where it is transcribed to produce primary piRNA transcripts that are processed by and bound to Piwi4, and transported to the cytoplasm where they participate in the targeting of viral RNA.

DOI: https://doi.org/10.7554/eLife.41244.011

transposon sequences (*Figure 5C*). We found more than 200 hybrids between SINV and the most abundant LTR-retrotransposon sequences (*Figures 5E* and 70 with Ty3 and 132 with Ty1 elements). Furthermore, mapping of hybrid reads from the most prevalent recombinants (SINV-Ty3/gypsy Element73) showed that recombination occurred in the virus 5'UTR, the non-structural viral genes and in the subgenomic viral RNA that contains the structural genes (*Figure 5F,i and ii*). In combination with the broad coverage of the SINV genome by the other SINV-only derived vDNA fragments (*Figure 5D,i*), our data suggest little bias in the recombination between LTR-retrotransposons and SINV mRNA. Similarly, CFAV-only and CFAV-LTR transposon hybrids reads (*Figure 5—figure supplement 1C*) appeared to originate and cover almost the entire viral genome. This further suggests that recombination is not driven by sequence homology. Unlike SINV, CFAV recombinants were found mostly associated with Ty1 elements (*Figure 5—figure supplement 1C*). We also found direct evidence of viral genome fragment conversion into circular vDNA (*Figure 5—figure supplement 1Di* and *ii*) in DENV infected Aag2 cells.

Next, we examined whether vDNA synthesis and recombination with transposon elements can discriminate between self (host) and nonself (viral) RNAs. Only a small fraction of known expressed Aag2 mRNAs were found to have derived sequences in the episomal DNAs reads (~7.5% of known expressed mRNAs; 1093 out of 14612). Furthermore, there was no correlation between mRNA and episomal DNA abundance (R = 0.0045, *Figure 5G* and *Figure 5—figure supplement 1E*). We detected no reads derived from the top 54 most abundant mRNAs, which account for 32% of all transcripts, and found no correlation between mRNA length and their enrichment in episomal DNA (R = 0.0786, *Figure 5H*). Indeed, the number of SINV vDNA reads was higher than for the most abundant host mRNA-derived episomal DNA (586 and 256 reads, for SINV and Aag2 transcript AAEL006357, respectively). Our results indicate that reverse transcription and recombination with retrotransposons is a highly specific mechanism, which preferentially converts viral RNA into vDNA. Together, these data suggest a model in which vDNA is acquired by the specific incorporation of viral RNA into retrotransposon replication complexes, followed by reverse transcription and recombination between the transposon and virus genome.

## Genomic analysis of Aag2 cells reveals independent acquisition of EVEs

If indeed reverse transcription and recombination are followed by integration into the host genome, this process could provide mosquitoes with a mechanism of immunological memory. Integration of vDNA derived from non-retroviral RNA viruses, known as endogenous viral elements (EVEs), have been identified in a wide variety of animal genomes (*Horie et al., 2010*; *Fort et al., 2012*; *Taylor et al., 2010*) but the function, if any, of EVEs is still ill-understood. We considered that EVEs in the *Ae. aegypti* genome (*Fort et al., 2012*; *Crochu et al., 2004*; *Katzourakis and Gifford, 2010*) may constitute a reservoir of virus sequences that can be utilized for the biogenesis of antiviral piRNAs. To examine this possibility, we sequenced the *Ae. aegypti* cell line Aag2 genome (~1.7 Gb) (*Whitfield et al., 2017*. The *Ae. aegypti* genome includes extreme heterozygosity, high repetitive content, and polymorphic inversions that together represent a significant challenge, particularly for the identification of newly acquired EVE sequences. We thus produced a high-quality assembly of the mosquito genome using single-molecule, real-time sequence technology (Pacific Biosciences sequencing) to generate long reads that permit the identification and analysis of highly repetitive regions (see Materials and methods). Our data enabled an assembly that was over 10 fold more contiguous than the previous *Aedes* assembly (*Whitfield et al., 2017*, and permitted examination of highly repetitive regions. This was important because virus/retrotransposon hybrids are likely to insert into the genome at target sites to form repetitive transposon clusters (*Bushman, 2003*. Thus, analysis of our assembly led to the identification of new EVEs in the Aag2 genome.

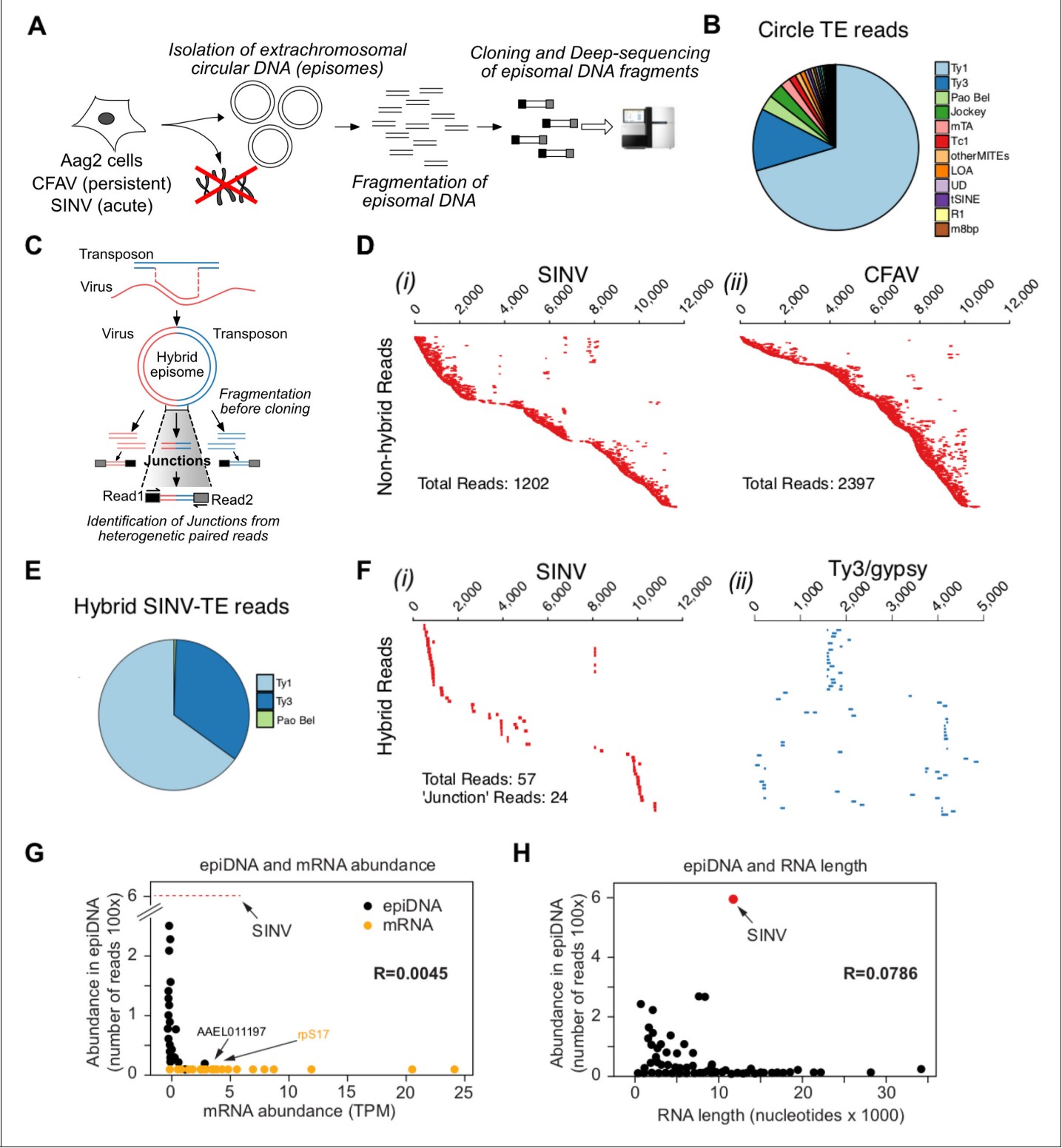

**Figure 5.** Conversion of viral RNA genome into viral DNA by recombination with retrotransposons is a discriminatory process.  (A Protocol for circular DNA sequencing extracted from infected cells. DNA was extracted from virus-infected Aag2 cells, circular DNA was purified by standard plasmid isolation protocol, treated with exonucleases and cloned for deep-sequencing using Nextera Illumina protocol. (B) Distribution of transposon (TE)-mapping reads from circular DNA sequencing. Of the 12 most abundant transposon families, all were retrotransposons except Tc1 and MITES/m8bp. (C Schematic representation of potential LTR-transposon/virus recombinants and the expected result in the circular DNA population. If virus/

*Figure 5 continued on next page*

*Figure 5 continued*

transposon recombinants accumulate during infection, hybrid reads should be present. In this example, paired-end read sequences should reveal viral sequences from read one and transposon sequence from read 2. D) All non-hybrid paired-end reads corresponding to SINV (i) or CFAV (ii) are represented by dashes aligned with respect to the corresponding genome (ordered vertically by most 5'-end sequence). Total number of reads observed varied from 1202 to 2397 depending on the virus. Length of dashes corresponds to length of read. Paired reads are shown on the same line. The respective genome lengths of SINV and CFAV are 11,703 and 10,695 nt. E Distribution of SINV-transposon hybrids reads from circular DNA sequencing. All transposon elements recombined with SINV sequences belong to the 3 families of LTR transposons. F Hybrid paired-end reads between SINV and Ty3/gypsy element 73 in SINV infected Aag2 cells. Viral-mapping (blue dashes i), and transposon-mapping (red dashes, ii) from paired read sequences are shown on the same line (across both virus and transposon). Reads are ordered based on the alignment to the CFAV viral genome starting with the most 5'-end sequence. 'Junction' reads refer to any paired-end reads where at least one read contained both virus and transposon sequence. G) Distribution of mRNA expressed in Aag2 cells based on their relative expression levels (in Transcripts Per Million, TPM) and their abundance in episomal DNA (in Nextera read counts). Aag2 mRNA with or without sequences identified in episomal DNA are shown in black and yellow, respectively. SINV abundance in episomal DNA is shown as a red line. The correlation coefficient R is shown. (H Distribution of mRNA expressed in Aag2 cells based on their length and abundance in episomal DNA (in Nextera read counts). The correlation coefficient R is shown.
DOI: https://doi.org/10.7554/eLife.41244.012

The following source data and figure supplement are available for figure 5:

**Source data 1.** Identification of virus-derived episomal DNAs in Sindbis or Dengue virus infected Aag2 cells.
DOI: https://doi.org/10.7554/eLife.41244.014

**Figure supplement 1.** Analysis of episomal viral and transposon-derived DNA in arbovirus-infected Aag2 cells.
DOI: https://doi.org/10.7554/eLife.41244.013

Among the EVEs identified in our new Aag2 genome (*Whitfield et al., 2017*, we found a CFAV-derived EVE within contig 000933F. This fragment corresponds to the CFAV NS2A gene coding region and it is flanked by long-terminal repeats (LTR) and *gag* Ty3-gypsy retrotransposon sequences (*Figure 6A*). This CFAV/Ty3 gypsy sequence is inserted into a piRNA producing cluster (piRNA cluster) among other LTR retrotransposons and EVEs. This piRNA cluster is transcriptionally active, producing antisense piRNAs targeting CFAV and members of other viral families, such as Rhabdoviridae and Reoviridae. Comparing this locus with the corresponding site in the *Ae. aegypti* Liverpool genome, from a strain isolated in West Africa, revealed that the CFAV EVE insertion observed in Aag2 cells is not present in the mosquito genome (*Figure 6B*), although other CFAV-derived EVEs are found at other loci in the *Ae. aegypti* AaegL3 assembly (*Palatini et al., 2017*. Of note, the Aag2 cell line was originally established in 1968 in Israel from mosquito embryos (*Peleg, 1968*. Thus, these observations suggest that acquisition of EVEs is a dynamic process, perhaps depending upon the geographic location and infection history of mosquito populations (*Palatini et al., 2017*; *Olson and Bonizzoni, 2017*.

## Piwi4 binds specifically to antiviral piRNAs produced from EVEs

To test if EVEs are used to produce functional antiviral piRNAs, we analyzed EVE-derived small RNAs produced in Aag2 cells. The vast majority of small RNAs mapping to EVEs are 24–30 nt in length (>95%) and are almost exclusively antisense relative to the pseudo-ORF (>99%) (*Figure 6C,i*). Furthermore, EVE-derived piRNAs showed a strong bias for uracil at the 1st position (*Figure 6—figure supplement 1A*) and were protected from beta-elimination, suggesting that they are mature piRNAs, methylated on their 3'end (*Figure 6C,i*). Importantly, knockdown of Piwi4 and Ago3, but not Ago2, resulted in a loss of EVE-derived piRNAs (*Figure 6C,ii*). Thus, the length, methylation, and sequence bias indicate that small RNAs derived from EVEs are *bona fide* piRNAs and their production requires Piwi4 and Ago3. Because Piwi4 specifically binds to antisense v-piRNAs produced during acute infection (*Figure 3A and C*), we examined whether Piwi4 also preferentially associates with antisense piRNAs during persistent infection. Analysis of the new, highly contiguous, Aag2 genome assembly revealed the existence of a unique CFAV-EVE in contig 933F, that produces high level of antisense piRNAs (*Figure 6A and B*) and corresponds to the NS2a gene of CFAV strain distinct from the CFAV that established persistent infection in the Aag2 cell line (see Methods). Accordingly, the sequence differences between CFAV-EVEs and CFAV genomic RNA allowed us to distinguish between EVE- and virus-derived piRNAs. Strikingly, Piwi4-associated small RNAs mostly corresponded to CFAV EVE-derived antisense piRNAs and not to CFAV virus-derived piRNAs (*Figure 6D* and *Figure 6—figure supplement 1B*) nor to TE-targeting piRNAs (*Figure 6—figure supplement*

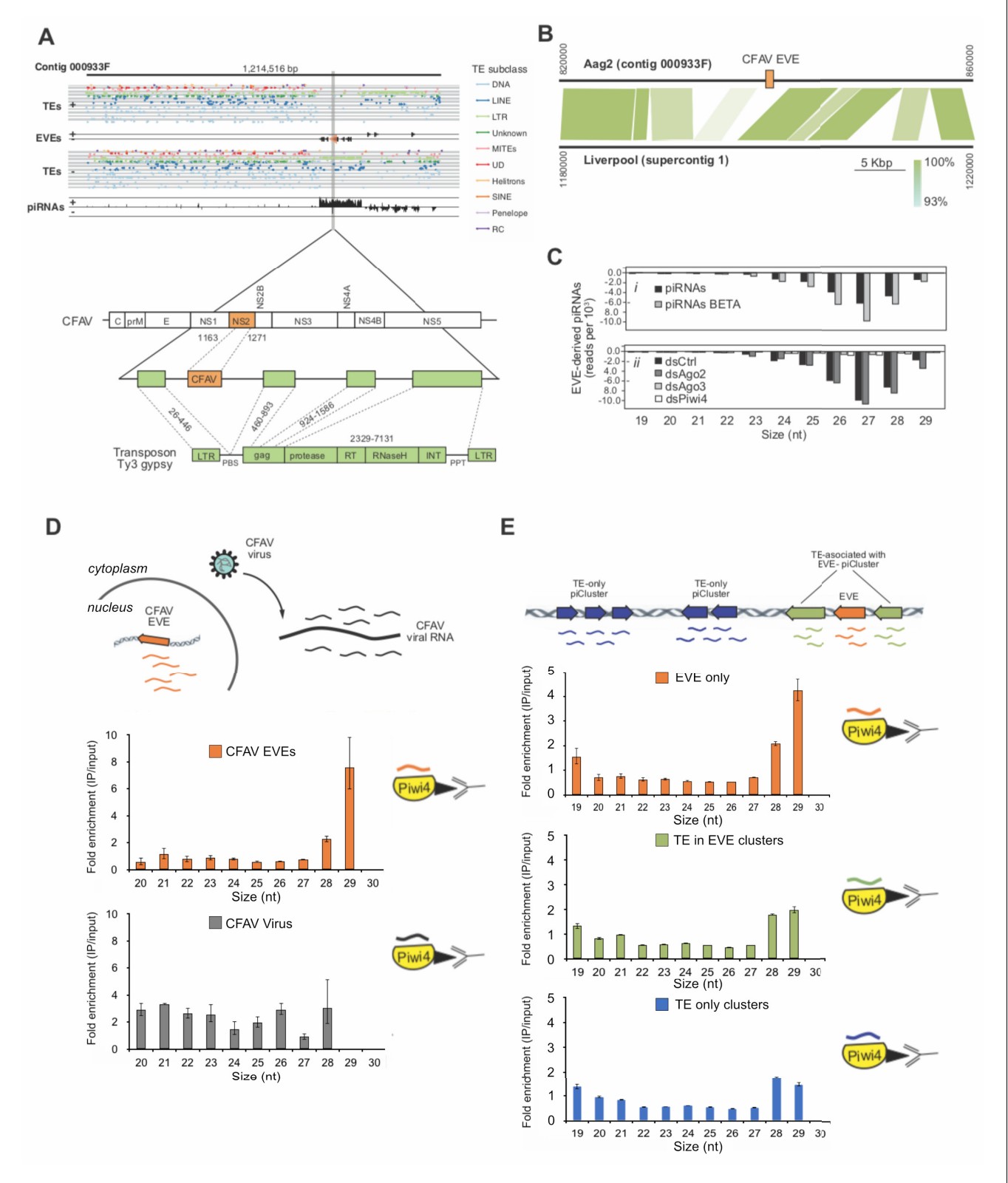

**Figure 6.** Acquisition of a CFAV EVE in the Aag2 genome and specific association of CFAV EVE-piRNAs with Piwi4. **A** (Upper panel) Distribution of transposons (TEs) and CFAV-derived EVE sequences and their respective mapping piRNAs and orientation within the genome (sense +, antisense -) in contig 933F from Aag2 genome. (Lower panel) Organization of the piRNA cluster containing the endogenized CFAV genome region (NS2) within Ty3/gypsy sequences. **B** Comparison of contig 933F in Aag2 genome and supercontig one in Liverpool mosquito genome. CFAV-derived EVE (orange) is

*Figure 6 continued on next page*

*Figure 6 continued*

only present in the interspersed transposon (green) sequences in the Aag2 genome. Sequence identity between the two contigs is expressed by color intensity according to the green scale. (C) Characterization of EVE-derived piRNAs (i) Size distribution plot of small RNAs mapping to the EVEs from Aag2 cells with or without β-elimination treatment. (ii) Size distribution plot of small RNAs mapping to the EVEs from Aag2 cells treated with control, Ago2, Piwi4, or Ago3 dsRNA. (D) Enrichment of antisense CFAV EVE (upper panel) or virus (lower panel)-derived small RNA in Piwi4-immunoprecipitation (IP) fraction compared to input sample. (E) Enrichment of genome-wide antisense EVEs (upper panel), TEs associated with EVE-piRNA clusters (middle panel) or TE-only piRNA clusters (lower panel) -derived small RNA in Piwi4-immunoprecipitation (IP) fraction compared to input sample.

DOI: https://doi.org/10.7554/eLife.41244.015

The following figure supplement is available for figure 6:

**Figure supplement 1.** Piwi4 binds to bona-fide antisense v-piRNAs produced by genomic CFAV-EVEs.

DOI: https://doi.org/10.7554/eLife.41244.016

*1C*). Although most piRNAs are between 26 and 28 nt long, Piwi4 associated with significantly longer piRNAs (28–29 nt vs 26–27 nt, $X\chi^2$, p<2.2e-16). Of note, during acute infection SINV-derived piRNAs bound to Piwi4 were also significantly enriched for longer sequences (*Figure 3A*). Again, the preferential binding of Piwi4 for longer piRNAs derived from genomic (EVE) versus RNA viral template was further confirmed in the other two replicates of the Piwi4-IP experiment (*Figure 6—figure supplement 1D*). Similar results were obtained with the majority of piRNAs derived from EVEs identified in the Aag2 genome (*Figure 6E*) demonstrating that Piwi4 preferentially associated with longer EVE-piRNAs at the genome-wide level (28–29 nt vs 26–27 nt, $^2$, p<2.2e-16).

The lack of enrichment in Piwi4 IP for the overall population of TE-targeting piRNAs (*Figure 6—figure supplement 1C*) suggests that Piwi4 specifically discriminates between TE-targeting and EVE-derived piRNAs. We hypothesized that the signals allowing this specific piRNA sorting to Piwi4 might be contained in the genomic loci of EVEs. Because EVEs are strongly enriched in a sub-population of TE-rich piRNA clusters interspersed with LTR TE fragments (*Whitfield et al., 2017*), loci-specific sorting to Piwi4 may then extend to neighboring TE-targeting piRNAs (which would have been masked in the genome-wide analysis). We thus examined whether TE-targeting piRNAs transcribed from piRNA clusters associated with EVEs were preferentially bound to Piwi4 compared to TE-targeting piRNAs derived from piRNA clusters that contained only TE fragments. Like for piRNAs derived from TE-only loci, analysis of TE-targeting piRNAs transcribed from EVE containing piRNA clusters showed no enrichment for Piwi4 binding (*Figure 6E*). Therefore, piRNAs that associate with Piwi4 does not appear to be specified at the global piRNA cluster level. Taken together, these results demonstrate that Piwi4 binds preferentially to longer piRNAs and distinguishes between antiviral EVE-derived piRNAs and other piRNA cluster TE-targeting piRNAs through localized EVE-specific signals.

## EVE-derived piRNAs and Piwi4 provide antiviral defense against acute and persistent infections

Like antisense v-piRNAs from acute SINV infection in Aag2 cells, CFAV EVE-derived piRNAs are efficiently loaded onto Piwi4, which acts as a restriction factor against acute arboviral infections (*Figure 1A and C*). Analysis of sense/anti-sense CFAV piRNAs showed a characteristic ping-pong processing, 10 bp offset signature (*Figure 7—figure supplement 1A*), suggesting that persistently infecting CFAV genome is cleaved by these anti-sense piRNAs (*Figure 7—figure supplement 1B*). We thus sought to determine whether piRNAs derived from EVEs are capable of mediating silencing. To this end, we inserted into the 3' UTR of a Renilla luciferase (Rluc) reporter gene ~500 bp sequences corresponding to *Ae. aegypti* genome-encoded EVEs in either the sense or antisense orientation (*Figure 7A,ii* inset). We selected four independent sequences with distinct piRNA expression levels (*Figure 7A,i* inset). Transfection of Rluc-EVE reporters into Aag2 cells demonstrated that Rluc expression was significantly reduced when the inserted EVE sequence was in the sense orientation as compared to the control antisense orientation (*Figure 7A,ii*). Furthermore, efficiency of silencing correlated with piRNA expression levels (*Figure 7A,i*). Thus, antisense EVE-derived piRNAs can silence mRNAs containing complementary sequences.

We next tested whether EVE-derived piRNAs can protect from virus infection. We first inserted two EVE sequences (derived from Wutai mosquito virus and CFAV) in the 3'UTR of SINV genome

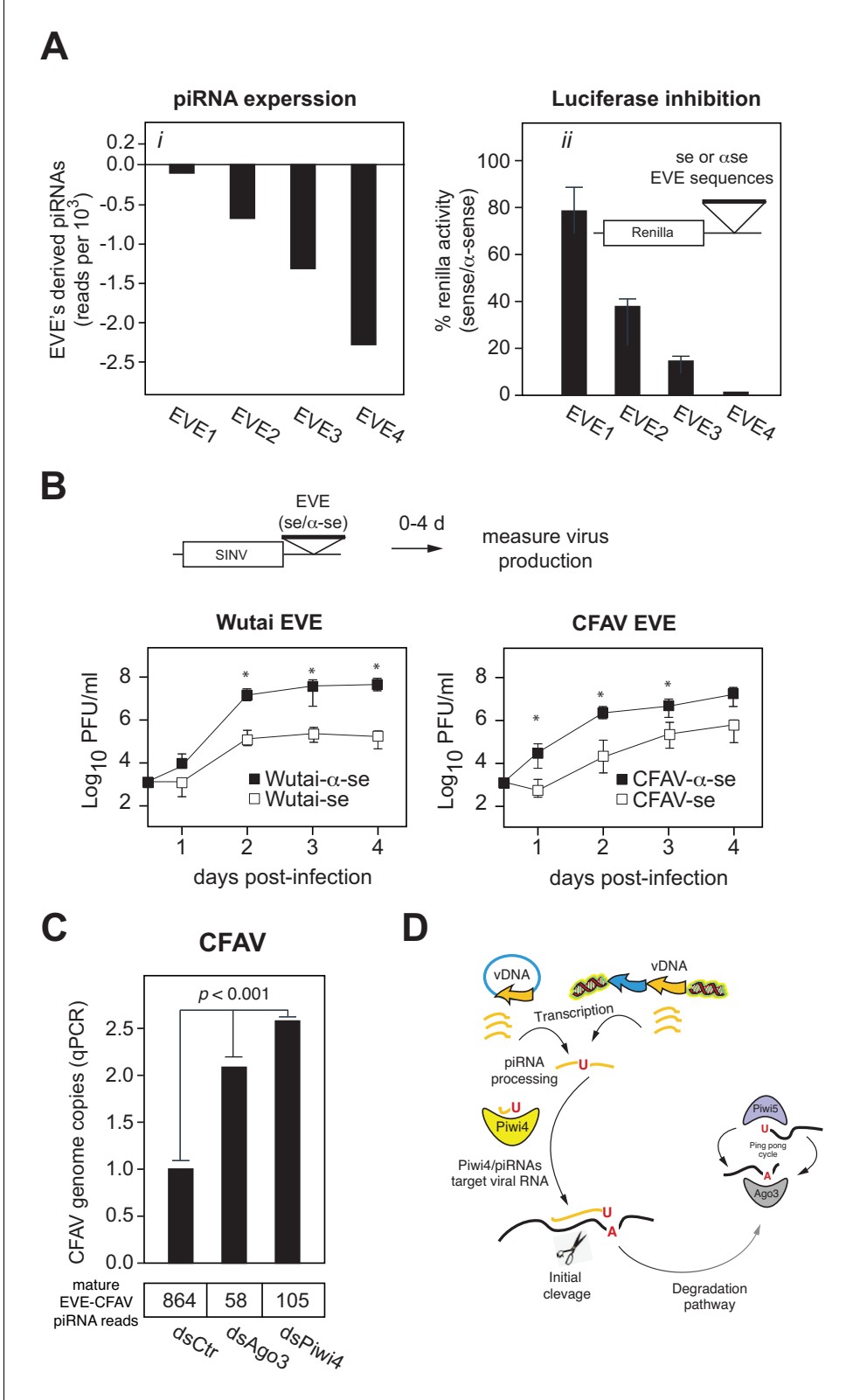

**Figure 7.** EVEs produce functional antiviral piRNAs that regulate virus attenuation. (**A**) (**i**) Abundance of piRNA reads (24-30nt long) derived from four distinct EVEs compared to the total number of small RNAs isolated from Aag2 cells (in reads per thousand). (**ii**) EVE-piRNA silencing reporters were designed by cloning a ~ 500 bp sense or antisense sequence from the identified piRNA producing EVEs in (**i**) into the 3' UTR of a Renilla luciferase expression vector (see schematic in inset). Silencing activity was measured as the ratio of Renilla luciferase activities from sense versus antisense EVE

*Figure 7 continued on next page*

*Figure 7 continued*

targets (normalized to firefly luciferase activity). (**B**) EVE-piRNA antiviral activity was assessed by infecting Aag2 cells with four modified SINV strains containing a ~ 350 nt sense or antisense sequence from identified piRNA-producing EVEs, targeting either Wutai Mosquito virus (left panel) or CFAV (right panel). Replication of SINV strains with a sense target to the endogenously expressed cognate piRNAs was significantly reduced compared to their antisense counterparts (asterisks correspond to p<0.05, one tailed Mann-Whitney-Wilcoxon test, n = 3 independent experiments). (**C**) Aag2 were transfected with dsRNA control (dsCtr), or directed against Ago3 (dsAgo3) or Piwi4 (dsPiwi4) and copies of CFAV genome per cell were determined by qPCR. Bottom table shows the number of mature piRNAs (i.e.: beta-elimination resistant) per condition in reads per millions.
DOI: https://doi.org/10.7554/eLife.41244.017

The following figure supplements are available for figure 7:

**Figure supplement 1.** piRNA mapping to CFAV viral genome and EVEs reveals multiple signatures of CFAV virus targeting by EVE-derived piRNAs.
DOI: https://doi.org/10.7554/eLife.41244.018

**Figure supplement 2.** EVE-piRNA antiviral activity is mediated by Piwi4.
DOI: https://doi.org/10.7554/eLife.41244.019

(*Figure 7B*). Infection of Aag2 cells with these viruses showed that insertion of EVE sequences in the sense orientation reduces virus replication 10 to 100 fold relative to the antisense EVE control viruses (*Figure 7B*, p<0.05 one-tail Wilcoxon rank sum test). We then assessed the role of Piwi4 in EVE-derived piRNA antiviral activity. Aag2 cells, pretreated with dsRNA against Piwi4 or eGFP as control, were infected with a SINV strain containing the Firefly luciferase reporter gene and a sense (S) or antisense (AS) EVE target sequence. Specific inhibition of the sense EVE target containing virus was assessed as the luciferase activity ratio of the SINV with the AS EVE over S EVE. In control cells (eGFP dsRNA), the relative viral replication advantage of the AS EVE containing virus over its S EVE counterparts increased over a 4 day time course. On the contrary, this ratio was significantly reduced at 48, 72 and 96 hr post infection (hpi) in Piwi4 knock-down Aag2 cells (*Figure 7—figure supplement 2*, Wilcoxon Signed Rank test, n = 4 independent experiments), indicating that Piwi4 is involved in the specific inhibition of virus replication by EVE-derived piRNAs in acute infection. We note that, while piRNAs that target the viral genome significantly decreased virus replication, it did not completely inhibit virus production. It is possible that the interaction of the EVE/piRNA system with the virus replication machinery may facilitate the establishment of a host-virus equilibrium typical of persistent infections. To test this hypothesis, we first determined the effect of Piwi4 knock-down on CFAV persistent infection and observed a significant increase in viral genome copy number (*Figure 7C*, p<0.001). Given that Piwi4 knockdown appears to affect both siRNA and piRNA production (*Figure 2C and D*), we sought to further examine the antiviral effect specific to piRNAs. To this end, we also performed Ago3 knockdown, which inhibits the production of piRNAs without significantly affecting siRNA levels (*Figures 2C* and *7C*, bottom panel). Ago3 knockdown also increased CFAV genome copy number (*Figure 7C*, p<0.001). We thus conclude that mosquito EVEs generate piRNAs capable of effectively controlling virus replication.

## Discussion

The immune system of *Ae. aegypti* mosquitoes allows arboviruses to establish persistent infection that facilitates spread in the human population. Investigating the role of the PIWI proteins in the mosquito immune response, we found that the anti-transposon surveillance pathway common to the majority of metazoan has evolved in mosquitoes as an effective adaptive antiviral defense. A central player in this pathway is Piwi4, which appears to specifically act in the mosquito antiviral piRNA system. In addition, episomal DNA and genome resequencing have revealed that conversion of viral RNA genomes into vDNA is a self-nonself discriminatory process and that EVE-derived piRNAs are loaded onto Piwi4 to restrict acute and persistent infections in mosquito cells.

### The antiviral factor Piwi4 binds to v-piRNAs and is required for their maturation

RNAi is the main antiviral defense in insects. Unlike other diptera such as *D. melanogaster*, arboviral infection of mosquito cells is known to lead to virus-derived piRNAs production (*Blair and Olson, 2015*), which is usually associated with genome surveillance against TEs (*Brennecke et al., 2007*). In addition, expansion of the Piwi gene family in *Ae. aegypti* suggested functional diversification of its

piRNA pathway. Indeed, studies of mosquito Piwi proteins have showed that Piwi5 and Ago3 bind to v-piRNAs and are involved in v-piRNA production but do not have antiviral activity. By contrast, Piwi4 was reported to control arboviral infections but not to bind to v-piRNAs nor contribute to their production. This discrepancy between antiviral function and piRNA production and binding has left our understanding of the piRNA machinery's role in antiviral defense incomplete. Accounting for viral RNA level and piRNA methylation, we now demonstrate that in fact, Piwi4 binds to antisense v-piRNAs and is required for their production and maturation. In this context, the lack of antiviral effect in Piwi5 and Ago3 knock down experiments (*Miesen et al., 2015*) suggests that v-piRNA accumulation from the ping pong amplification between Piwi5 and Ago3 might solely reflect a degradation pathway for viral RNAs already silenced by other RNAi complexes, such as Piwi4 and Ago2.

In SINV-infected Aag2 cells, knock down of Ago2, which binds to v-siRNAs, led to a decrease of mature v-siRNAs but not of mature v-piRNA production. By contrast, knock down of Piwi4, which we found binds to v-piRNAs, led to a loss of both mature v-siRNAs and v-piRNAs. In addition, Piwi4 does not bind to TE-targeting piRNAs (*Figure 3B*) but is required for their maturation (*Figure 2— figure supplement 1F*). Finally, our results and others (*Varjak et al., 2017a*) indicate that Piwi4 and Ago2 interact. Together, these data suggest that Piwi4 acts as a hub between the siRNA and piRNA antiviral pathways in mosquito and could affect siRNAs and piRNAs maturation by controlling the recruitment of the endogenous small RNA methyltransferase to RISCs. Ago3, which is also involved in TE targeting and v-piRNA methylation does not affect v-siRNA maturation and thus does not share the central role of Piwi4.

## Piwi4 binds to a specific form of v-piRNAs transcribed from vDNA elements

Production of vDNA molecules in *Aedes* mosquitoes during arboviral infection is linked to viral tolerance in mosquito vector (*Goic et al., 2016*) and consequently a capacity to act as an arbovirus vector. By sequencing episomal DNA in SINV and DENV infected Aag2 cells, we have confirmed that fragments of the infecting arbovirus are reverse-transcribed with LTR TEs providing unbiased coverage from loci across the entire viral RNA genome. Thus, recombination between viral genomes and transposons appears to be sequence-independent. Importantly, our analysis reveals that vDNA formation occurs through a self-nonself discriminatory process. An additional characteristic of a *bona fide* immune response to viral infection.

In the genome, vDNA sequences known as EVEs produce antisense v-piRNAs that can restrict acute and persistent infections (*Figure 7*). During processing, Piwi4 binds specifically to antisense v-piRNAs produced during acute SINV infection and EVE-derived piRNAs. In both cases, Piwi4 associates preferentially with long forms of v-piRNAs (28–29 nucleotides in length). Furthermore, inhibition of vDNA production leads to increased viral replication and decreased v-piRNA accumulation during acute SINV infection, similar to Piwi4 knock down. Because of this phenocopy between vDNA inhibition and Piwi4 knock down and the conserved binding preference of Piwi4 to v-piRNAs derived from EVEs and acute infection, we propose that antisense v-piRNAs bound to Piwi4 during acute arborviral infection derived from vDNA transcription rather than from the viral RNA itself. Consistent with this model, the lack of effect of Ago2 knock down on v-piRNA production despite Ago2 interaction with Piwi4, further suggests that v-piRNAs bound to Piwi4 are not processed from viral RNA genomes cleaved by Ago2-RISCs.

In Aag2 cells, EVE-containing piRNA clusters produce more piRNAs than loci without EVEs (*Whitfield et al., 2017*). However, little is known about the regulatory mechanisms behind EVE-derived piRNAs production. Preferential binding of Piwi4 to EVE-derived v-piRNAs does not extend to neighboring TE-targeting piRNAs (*Figure 6E*). Therefore, sorting of EVE-piRNAs to Piwi4 might be regulated in a target-specific manner, that is able to discriminate between TE and virus sequences at the genomic level. In *Drosophila*, heterochromatin-dependent recruitment of the transcription machinery allows internal initiation within piRNA clusters and specific transcription of TE-targeting piRNA clusters versus TEs (*Andersen et al., 2017*). Similarly, we speculate that epigenetic marks on EVEs might control their transcription and the specific sorting of EVE-derived piRNAs to Piwi4.

# Endogenized viral elements in the mosquito genome provide transgenerational antiviral immunity

Our study suggests that in *Aedes* mosquitoes, the piRNA pathway has acquired a novel antiviral function with specialized Piwi proteins, such as Piwi4, to restrict viral replication in somatic tissues of infected mosquitoes. Antiviral piRNA biogenesis is initiated by the formation of EVEs that is dependent on RT-mediated vDNA synthesis and integration into the genome (*Figures 4B* and *6B* and C). EVEs derived from vertically transmitted insect-specific viruses (ISV) (*Whitfield et al., 2017*; *Palatini et al., 2017*) are prevalent in the mosquito genome. Indeed, germline infection by ISVs could facilitate EVE endogenization and vertical transmission. Phylogenetic and phenotypic analysis of ISVs and arboviruses from the Bunyaviridae and Flaviviridae families suggests an arthropod origin to arboviruses (*Marklewitz et al., 2015*). In addition, ISV persistent infections in mosquitoes can restrict replication of some arboviruses (*Hall et al., 2016*). Our results indicate that EVE-derived piRNAs decrease the replication of viruses with complementary sequences. In this context, we speculate that EVE-derived RNAi immunity against ISVs could force them away from the germline and drive a change from vertical to horizontal transmission leading to the emergence of arboviruses.

Conversely, for arboviruses that are already in circulation in vertebrate hosts and do not infect the mosquito germ line, additional mechanisms must exit to ensure maintenance of viral tolerance in the mosquito vector. Our work in adult female *Ae. aegypti* shows that Piwi4 is upregulated in the midguts and carcasses after blood meal but not after sugar meal, suggesting that Piwi4 expression is stimulated in conditions associated with arbovirus exposure in vivo. Also, Piwi4 binds specifically to antisense v-piRNAs derived from vDNA elements (*Figure 6D*). In addition, inhibition of vDNA formation in vivo leads to *Ae. Aegypti* hypersensitivity to CHKV infection (*Goic et al., 2016*). Together, these results suggest that exposure to circulating arboviruses could trigger an antiviral piRNA response in somatic tissues, consistent with the establishment of arbovirus persistent infection.

Finally, maternal inheritance of cytoplasmic piRNAs from active piRNA loci can convert homologous inactive clusters into strong piRNA producing loci in *Drosophila* (*de Vanssay et al., 2012*). Similar cross-talk between EVEs and newly acquired vDNA from infecting arboviruses with partial identity to EVEs could suffice to trigger piRNA production from vDNA and thus stimulate efficient antiviral activity through v-piRNAs and Piwi4. We propose that EVEs and Piwi4 are functional extensions of the piRNA pathway and provide a conserved mechanism of transgenerational adaptive antiviral immunity in mosquito.

# Materials and methods

### Key resources table

| Reagent type (species) or resource | Designation | Source or reference | Identifiers | Additional information |
|---|---|---|---|---|
| Gene (*Aedes aegypti*) | *Ago3* | https://www.vectorbase.org | | Previously known as LOC5569680 |
| Gene (*Aedes aegypti*) | *Piwi4* | https://www.vectorbase.org | | Previously known as AAEL007698 |
| Gene (*Aedes aegypti*) | *Ago2* | https://www.vectorbase.org | | Previously known as AAEL017251 |
| Strain, strain background (Sindbis virus) | Sindbis virus | *Strauss et al., 1984* | | Sindbis virus was produced and titered in BHK-21 cells. |
| Strain, strain background (Dengue virus) | Dengue Virus type 2 Thailand 16681 | *Kinney et al., 1997* | | |

*Continued on next page*

*Continued*

| Reagent type (species) or resource | Designation | Source or reference | Identifiers | Additional information |
|---|---|---|---|---|
| Strain, strain background (Dengue virus) | Dengue Virus type 2 Jamaica 1409 | *Pierro et al., 2006* | | |
| Strain, strain background (*Aedes aegypti*) | Chetumal strain | NA | | *Ae. aegypti* female mosquitoes were reared, infected and injected in the Olson lab |
| Transfected construct (*Ae. Aegypt*) | pAcFLuc | This paper | | *Ae. Aegypti* poly-ubiquitin promoter driving expression of Firefly Luciferase |
| Transfected construct (*D. melanogaster*) | pUbRLuc | This paper | | *D. melanogaster* Actin promoter driving expression of Renilla Luciferase |
| Transfected construct (*Ae. Aegypi*) | pUb:FLAG-Piwi4 | This paper | | *Ae. Aegypti* poly-ubiquitin promoter driving expression of a FLAGx3-tagged Piwi4 coding sequence |
| Antibody | anti-FLAG M2 Magnetic beads (mouse monoclonal) | Sigma-Aldrich | M8823 | |
| Recombinant DNA reagent | SINV:Fluc-EVE-Sense | This paper | | Sindbis virus strain modified to include an additional subgenomic promoter to express Firefly luciferase with a 185nt sense Xincheng mosquito virus glycoprotein EVE fragment in its 3'UTR. |
| Recombinant DNA reagent | SINV:Fluc-EVE-AntiSense | This paper | | Sindbis virus strain modified to include an additional subgenomic promoter to express Firefly luciferase with the 185nt sense Xincheng mosquito virus glycoprotein EVE fragment in the antisense orientation in its 3'UTR. |

*Continued on next page*

*Continued*

| Reagent type (species) or resource | Designation | Source or reference | Identifiers | Additional information |
|---|---|---|---|---|
| Recombinant DNA reagent | SINV:Wutai-Sense | This paper | | Sindbis virus strain modified to include an additional subgenomic promoter followed by a 334nt sense Wutai nucleocapsid EVE fragment in its 3'UTR. |
| Recombinant DNA reagent | SINV:Wutai-AntiSense | This paper | | Sindbis virus strain modified to include an additional subgenomic promoter followed by the 334nt Wutai nucleocapsid EVE fragment in antisense orientation in its 3'UTR. |
| Recombinant DNA reagent | SINV:CFAV-Sense | This paper | | Sindbis virus strain modified to include an additional subgenomic promoter followed by a 326 nt sense CFAV NS2a EVE fragment in its 3'UTR. |
| Recombinant DNA reagent | SINV:CFAV-AntiSense | This paper | | Sindbis virus strain modified to include an additional subgenomic promoter followed by the 326nt CFAV NS2a EVE fragment in antisense orientation in its 3'UTR. |
| Commercial assay or kit | Luciferase Assay System | Promega | E1500 | |
| Commercial assay or kit | Dual-Glo Luciferase Assay System | Promega | E1910 | |
| Commercial assay or kit | NEBNext Small RNA Library Prep Set for Illumina | NEB | E7330S | |
| Chemical compound, drug | 3'-azido-3'-deoxythymidine (AZT) | Synthonix | A2698 | |
| Software, algorithm | Bowtie | *Langmead et al., 2009* | | http://bowtie-bio.sourceforge.net/index.shtml |
| Software, algorithm | Python (version 2.7.6) | Python Software Foundation,; 2017 | | https://www.python.org |

*Continued on next page*

*Continued*

| Reagent type (species) or resource | Designation | Source or reference | Identifiers | Additional information |
|---|---|---|---|---|
| Software, algorithm | R (version 3.30) | R Core TeamR: a language and environment for statistical computing. R Foundation for Statitstical Computing; 2014 | | https://r-project.org/ |
| Software, algorithm | FASTX Toolkit | Hannon lab | | http://hannonlab.cshl.edu/fastx_toolkit/ |
| Software, algorithm | WebLogo 3 | *Crooks et al., 2004* | | weblogo.three plusone.com |

## Cell culture and virus propagation

*Aedes aegypti* Aag2 cells were cultured at 28°C without $CO_2$ in Schneider's *Drosophila* medium (GIBCO-Invitrogen), supplemented with 10% heat-inactivated fetal bovine serum (FBS), 1% non-essential amino acids (NEAA, UCSF Cell Culture Facility, 0.1 µm filtered, 10 mM each of Glycine, L-Alanine, L-Asparagine, L-Aspartic acid, L-Glutamic Acid, L-Proline, and L-Serine in de-ionized water), and 1% Penicillin-Streptomycin-Glutamine (Pen/Strep/G, 10,000 units/ml of penicillin, 10,000 µg/ml of streptomycin, and 29.2 mg/ml of L- glutamine, Gibco).

SINV (J02363) and CHIKV (vaccine strain 181/25) stocks were produced by infecting Vero cells at low MOI (below 1) in MEM with 2% heat-inactivated FBS. After CPE was observed (~72 hr post infection), the supernatant was centrifuged at 3000 g for 10 min, passed through a 0.45 µm filter, supplemented with 10% glycerol, flash frozen, and stored at −80°C. Recombinant SINV-EVE strains were generated by inserting after an additional sub genomic promoter, a sequence cloned from a Wutai Mosquito virus-derived EVE (328 bp) or CFAV-derived EVE (332 bp) in the Aag2 cells genome. Cloning was performed using InFusion (Clontech) after PCR amplification of EVE regions in the sense (oligos IF5/IF6 for CFAV-EVE, IF9/IF10 for WutaiVirus-EVE) and antisense orientation (oligos IF7/IF8 for CFAV-EVE, IF11/IF12 for WutaiVirus-EVE).

IF5_AagEVE_CFAV-NS2a_F
ACCACCACCTCTAGATGGGTGTTGCTAGTGGCG
IF6_AagEVE_CFAV-NS2a_R GGATCCATGGTCTAGTGCGGCCGCTCTTCCACCCCATTATCAGGC
IF7_AagEVE_CFAV-NS2aAS_F
ACCACCACCTCTAGATCTTCCACCCCATTATCAGGC
IF8_AagEVE_CFAV-NS2aAS_R GGATCCATGGTCTAGTGCGGCCGCTGGGTGTTGCTAGTGGCG
IF9_AagEVE_WutaiV_F
ACCACCACCTCTAGACAATAATCTCAATGATGTCCTCGCG
IF10_AagEVE_WutaiV_R
GGATCCATGGTCTAGTGCGGCCGCCGCTATTGGACAGATTGTAGACTGT
IF11_AagEVE_WutaiV_AS_F
ACCACCACCTCTAGACGCTATTGGACAGATTGTAGACTGT
IF12_AagEVE_WutaiV_AS_R  GGATCCATGGTCTAGTGCGGCCGCCAATAATCTCAATGATGTCCTCGCG

Recombinant SINV:Fluc:EVE strains were generated by inserting in the 3' UTR of a Firefly *luciferase* gene, located downstream of an additional sub genomic promote (SINV:Fluc), a sequence cloned from a Xincheng Mosquito virus-derived EVE (185 bp) in the Aag2 cells genome (with an output of mature 995 antisense piRNAs per million reads). Cloning was performed using NEBuilder HiFi DNA Assembly (NEB) after PCR amplification of the EVE regions (oligos GA50/GA51 for the sense EVE and GA52/GA53 for the antisense EVE) and inserted into a PmeI digested pSINV:Fluc vector.

GA50: aaagatcgccgtgtgagttttgacccaaattccctcttaag
GA51: cgaggctgatcagcgggtttaatggaaatattaagaataatgaaattgatcc
GA52: aaagatcgccgtgtgagtttGAaatggaaatattaagaataatgaaattgatcc
GA53: cgaggctgatcagcgggtttgacccaaattccctcttaag

DENV2 (Thailand 16681 for cell culture experiments and Jamaica 1409 for in vivo experiments) stocks for cell culture studies were produced as described above except that Huh7 cells were used for propagation and the viral stock was supplemented with 20% FBS instead of glycerol. DENV2 for mosquito studies was propagated in C6/36 (*Ae. albopictus*) cells cultured in Modified Eagle's medium (MEM) supplemented with 7% heat inactivated (56°C for 30 min) fetal bovine serum (FBS), 1% penicillin/streptomycin, 1% glutamine, 1% NEAA and maintained at 28°C at 5% $CO_2$.

## Virus titration

Virus was titrated by plaque assay by infecting confluent monolayers of Vero cells with serial dilutions of virus. Cells were incubated under an agarose layer for 2 to 3 days at 37°C before being fixed in 2% formaldehyde and stained with crystal violet solution (0.2% crystal violet and 20% ethanol). DENV2 plaque assays were performed in LLC-MK2 cells in 24-well plates. Ten-fold serial dilutions of whole mosquito homogenate supernatant were added for 1 hr and cells were overlaid with agar. After 7 days of incubations at 37°C cells were stained by addition of 3 mg/ml MTT (3-[4,5-dimethylth-iazol-2-yl]−2,5-diphenyltetrazolium bromide) solution to the plate and incubated for 4 hr (*Sladowski et al., 1993*; *Takeuchi et al., 1991*). Viral titers were calculated, taking into account plaque numbers and the dilution factor.

## dsRNA preparation

PCR primers including the T7 RNA polymerase promoter were used to amplify in vitro templates for RNA synthesis using Phusion polymerase (NEB). Manufacturer's recommendations were used for the concentrations of all reagents in the PCR. Primers were synthesized by *Integrated DNA Technologies*, Inc (IDT). The thermocycling protocols are as follows: 98°C 2:00, (98°C 0:15, 65°C 0:15, 72°C 0:45, these three cycles were repeated 10X with a lowering of the annealing temperature by 1°C per cycle); (98°C 0:15, 60°C 0:15, 72°C 0:45, these three steps were repeated 30X), 72°C 2:00. RNA was synthesized in a 100 μl in vitro transcription (IVT) reaction containing 30 μl of PCR product, 20 μl 5X IVT buffer (400 mM HEPES, 120 mM $MgCl_2$, 10 mM Spermidine, 200 mM DTT), 16 μl 25 mM rNTPs, and 1 unit of T7 RNA polymerase. The IVT reaction was incubated at 37°C for 3–6 hr and then 1 μl of DNase-I (NEB) was added and the reaction was further incubated at 37°C for 30 min. The RNA was purified by phenol-chloroform-isoamyl alcohol followed by isopropanol precipitation. RNA was quantified using a Nanodrop (Thermo Scientific) and analyzed by agarose gel electrophoresis to ensure integrity and correct size.

## dsRNA soaking

Prior to dsRNA soaking, Aag2 cells were washed once with phosphate buffered saline w/o calcium or magnesium (dPBS, 0.1 μM filtered, 0.2 g/L $KH_2PO_4$, 2.16 g/L $Na_2HPO_4$, 0.2 g/L KCl, 8.0 g/L NaCl). Cells were soaked in 5 μg /ml dsRNA in minimal medium (Schneider's *Drosophila* medium, 0.5% FBS, 1% NEAA, and 1% Pen/Strep/G) for the time indicated by the experiment. All incubations were performed at 28°C without $CO_2$. 3 days later, dsRNA-treated Aag2 cells were infected with SINV (MOI = 1) and collected at 3 day post infection.

## Ae. aegypti

Adult *Ae. aegypti* (Chetumal strain) female mosquitoes were taken 2–3 days after emerging from the pupa stage and intrathoracically (IT) injected with 500 ng of dsRNA 2–4 days post-emergence. No randomization or blinding of the group allocation of mosquitoes was used for these studies. Three days later mosquitoes were infected with DENV-2 Jam1409 using an artificial bloodmeal consisting of DENV2-infected C6/36 (A. albopictus) cells in L15 medium suspension (60% vol), defibrinated sheep blood (40% vol); Colorado Serum Co., Boulder, CO) and 1 mM ATP. DENV2 titers in the bloodmeal were $1.0 \pm 0.4 \times 10^6$ PFU/mL. Blood fed females were selected, provided water/sugar and maintained in the insectary at 28°C, 82% relative humidity. For all mosquito experiments sample size was selected based on known variability from previous experiments.

Ovaries, midguts and carcasses were dissected at 0, 4, 7, 10 and 14 dpi and collected in 100 μl of Trizol reagent (Invitrogen) and total RNA was extracted following the manufacturer's instructions. Whole mosquitoes were also collected at 0, 4, 7, 10 and 14 dpi and triturated in 1 ml of

supplemented MEM. After centrifugation the supernatant fluid was filtered (Acrodisc Syringe filters with 0.2 µm HT Tuffryn membrane) and virus titers were determined by plaque assay.

## Identification of CFAV strains

Primers (CFAV-fw and CFAV-rev) were designed to anneal to regions flanking the NS2A coding sequence of CFAV that are conserved between the Bristol (Gen- Bank# KU936054) and the Culebra (GenBank# AH015271.2) strains (Region 3321–3851 in Bristol coordinates). Sanger sequencing of the amplicon produced with these primers from cDNA generated from Aag2 cells identified the persistent infecting CFAV virus as belonging to the Bristol strain.

CFAV-fw: GCGAGGAACCAGAACCAACA
CFAV-rev: GCAGGACGCTCTTGTAGGC

## Luciferase assays

Cells were soaked in dsRNA for the indicated period of time using the dsRNA soaking method above. Prior to transfection, cells were washed 3X with dPBS, and then added to complete medium. Cells were transfected with plasmids encoding Firefly (pAc Fluc) and Renilla luciferase (pUb Rluc) with Effectene (Qiagen) using the manufacturer's instructions with the following modifica- tion: 200 ng pAc Fluc and 50 ng pUb Rluc were used per transfection with a ratio of 1 µl effectene/250 ng plasmid DNA. Firefly and *Renilla luciferase* sequences from the plasmids pGL3 and pRL-CMV (Promega) were cloned into pAc/V5-HisB (Invitrogen) and pUb (Ubiquitin promoter; *Anderson et al., 2010*), respectively.

Twenty-four hours post transfection, cells were lysed in 50 µl passive lysis buffer (Promega), and Firefly and Renilla luciferase activity was determined from 10 µl of lysate using a dual luciferase reporter assay system using the manufacturer's instructions (Promega) and analyzed on an Ultra-evolution plate reader (Tecan) using an integration time of 100 ms.

The analysis of effects of dsRNA uptake on target gene candidates was performed as above with the following exceptions: 30,000 cells were seeded per well, cells were treated with the initial dsRNA (targeting the candidate gene) for 72 hr, washed 3X with dPBS, before the addition of the secondary dsRNA (targeting the reporter).

For the effect of Piwi4 knock-down on the silencing of EVE containing viruses (SINV:Fluc:EVE) (*Figure 7—figure supplement 2*), $25 \times 10^3$ Aag2 cells (per well in a 96-well plate format) were transfected (TransIT 20–20, Mirus) with 100 ng of dsRNA against eGFP or Piwi4. Two days later, cells were washed once with PBS and infected with $10^5$ pfu of SINV:Fluc:EVE Sense or Antisense in 50 µL minimal media for 1 hr at 28°C. Then, 100 µL complete media (10% FBS) with 20 ng dsRNA dsRNA against eGFP or Piwi4 according to the initial transfection, were added to each well. After 24, 48, 72 and 96 hpi, cells were rinsed with PBS, lysed for 15 min at room temperature in 30 µL 1x Passive Lysis Buffer (Promega). 15 µL of cell lysate were transferred to a 96-well white plate (Costar) and mixed with 75 µL Luciferase Activity Reagent (Promega) and analyzed on an Veritas plate reader.

## qPCR analysis

Total RNA was extracted using TRIzol (Life Technologies). cDNA synthesis was performed using the iScript cDNA synthesis kit (Bio-Rad). Primers for RT-qPCR were obtained from IDT and are listed in the supplementary material. Specific genes or viral genomes were analyzed using SYBR green methods on a CFX- Connect (Bio-Rad). All genes/viruses tested for relative quantitation were normalized to RP49 expression. Relative quantitation was calculated by the $2^{-ddCt}$ method. Absolute quantitation was calculated using a serial dilution of a plasmid containing the viral or host gene of interest.

## Statistical analysis

Values were expressed as means + /- standard deviation. All viral titrations and qRT-PCR experiments were performed on four independent samples. The distributions of RNA expression levels and viral titers were tested for normality using the Shapiro-Wilk test in R (shapiro.test()) and found to be not normal. Therefore all related statistical analyses were performed using the non-parametric two-tailed Mann-Whitney U test (wilcox.test() in R) to assess whether the distributions of control versus treated conditions were different. When they were, $p$ was always equal to 0.02857 due to the sample size and was reported as p<0.05.

Comparison of growth curves of Sindbis virus strains containing a sense versus antisense cognate sequence to endogenous EVE-derived piRNAs were performed on the mean values of three independent infection experiments using the non-parametric (no assumption of normal distribution) Mann-Whitney-Wilcoxon test. Viral strains with sense targets for antisense EVE-piRNAs were expected to have decreased titers and therefore one tailed tests were used.

Chi square ($X^2$) tests were performed on $2 \times 2$ contingency tables for the different size or strandness of v-piRNAs using the chisq.test in R.

## Affinity purification (AP)

five $\times\ 10^6$ Aag2 cells were seeded in 10 cm dishes and allowed to attach overnight. Cells were transfected with expression plasmids using Transit2020 (Mirius Bio) using the manufacturer's instructions. 24 hr post transfection cells were washed with dPBS three times, scraped off the dish in IP- buffer (pH 7.5 @ 4°C, 10 mM Tris, 2.5 mM EDTA, 250 mM NaCl, and cOmplete protease inhibitor, Roche), and centrifuged at 2000 rcf for 5 min at 4°C. Cell pellets were resuspended in 300 µl lysis buffer (IP-buffer + 0.5% NP-40) and incubated at 4°C for 30 min and then centrifuged at @ 12,000 rcf for 10 min at 4°C. The supernatant was added to 50 µl of Protein A conjugated beads (Sigma) and rotated for 1 hr at 4°C. Lysate was adjusted to 1% NP40 (by adding four volumes IP-buffer) and then transferred to 50 µl of anti-FLAG conjugated beads (Sigma, Cat #F2426) and rotated for 6–16 hr at 4°C. Beads were then washed six times with wash-buffer (IP-buffer + 0.05% NP- 40). Following the final wash 30 µl of elution buffer was added (IP-buffer + 100 µg/ml 3x Flag peptide, Sigma) and rotated for 1 hr at 4°C.

## Mass spectrometry

To ensure samples were appropriate for mass spectrometry, eluates from affinity purification were analyzed by western blot and silver stain (Pierce). Samples were run on an Orbitrap LC-MS and analyzed with MaxQuant software.

## Piwi4-associated piRNAs

Piwi4 Flag was overexpressed in Aag2 cell and immunoprecipitated from SINV infected Aag2 cells at three dpi. After three washes with TBS + 1 mM EDTA and 80 U/mL murine RNase Inhibitors (NEB), 1 ml of TRIzol was added to Flag beads and RNA extraction was performed according to the manufacturer's protocol. Cloning of small RNAs was performed as described below.

The additional Piwi4 pull-down experiments were performed similarly except for the following. After the three washes, small RNAs were first released from the beads by adding 20 mg/mL Proteinase Kand 80 U/mL murine RNase Inhibitors for 1 hr at 55°C, followed by TRIzol extraction. Cloning of the small RNAs were performed as described below except that there was no size selection for the bound fraction and ligation of the 3' adapter was performed overnight at 18°C for increase efficiency.

For the eGFP control pull-down experiment, a pUb-eGFP plasmid was transfected instead of the pUb-Piwi4:FLAG. Pull down and small RNA extraction and cloning were performed as described for the additional Piwi4 pull-down experiments.

## Small RNA cloning for deep sequencing

seven7 $\times\ 10^6$ Aag2 cells were seeded in each T-75 flask in complete medium and allowed to attach overnight. Cells were washed with dPBS three times, scraped off the dish in dPBS, and centrifuged at 2000 rcf for 5 min at 4°C. RNAs were isolated using the miRvana kit (Life technologies). The large RNA fraction was used for RT-qPCR. The small RNA fraction was precipitated by adding 1/10th volume 3M NaOaC pH 3.0, 1 µL gylcoblue (Life technologies), and 2.5 volumes 100% EtOH and incubated at −80°C at least 4 hr and then centrifuged at 12,000 rcf for 10 min at 4°C. The pellet was washed with 80% EtOH and then resuspended in Gel Loading Buffer II (Life Technologies) and run on a 16% polyacrylamide gel containing 8M urea. Small RNAs (17–30 nt) were cut out from the gel and eluted overnight at 4°C and precipitated by adding 1/10th volume 3M NaOaC pH 3.0, 1 µL gylcoblue (Life technologies), and 2.5 volumes 100% EtOH and incubated at −80°C at least 4 hr and then centrifuged at 12,000 rcf for 10 min at 4°C. Small RNAs were cloned using microRNA Cloning Linker1 (IDT) for the 3' ligation (ligation at 25°C for 3 hr) and the modified 5' adapter (ligation at 37°

C for 2 hr) with randomized 3'-end (CCTTGrGrCrArCrCrCrGrArGrGrArArTrTrCrCrArNrNrNrNrN). Libraries were run on a HiSeq 1500 or 2500 using the Rapid run protocol.

For circular DNA deep-sequencing, cytoplasmic fractions were isolated from Aag2 cells and processed for DNA extraction using a Nucleospin tissue kit (Macherey-Nigel). DNA was treated overnight with Plasmid-Safe ATP-dependent-DNase (Epicentre) overnight at 37°C to remove non-circular DNA. 1 ng of Plasmid-Safe treated DNA was used for Nextera cloning (Nextera DNA Library sample prep kit, Illumina) and sequenced on an HiSeq 4000. Bioinformatic analyses were carried out as described (*Tong et al., 2017*). More than 50% comes from the genome (but are not mitochondrial nor TE derived). Of that 50%, about 1% comes from transcribed regions, suggesting very little contamination from cellular transcripts.

## Small RNA bioinformatics

Adaptors were trimmed using Cutadapt (*Chen et al., 2014*) with the –discard-untrimmed and -m 19 flags to discard reads without adaptors and below 19 nt in length. Reads were mapped using bowtie (*Langmead et al., 2009*) using the –v one flag. Read distance overlaps were generated by viROME (*Watson et al., 2013*). Sequence biases were determined by Weblogo (*Crooks et al., 2004*). Transposon sequences for *Ae. aegypti* were downloaded from TEfam (https://tefam.biochem.vt.edu/tefam/browse.php).

Sequencing depth, virus and TE read counts are presented in the *Tables 1* and *2*.

For the Piwi4 pull-down experiments, small RNA reads were mapped using Bowtie 1.2.1.1 in v mode, allowing up to two mismatches (-v 2). SINV and transposon-derived reads were mapped using the SINV genome (accession # NC_001547) and the TEfam database respectively. To specifically identify small RNAs derived from CFAV-EVEs, reads were first filtered out by mapping them to the persistently infecting CFAV virus genome (Bristol strain, accession # KU936054). Unmapped reads were then aligned to CFAV-EVEs identified in the PacBio Aag2 genome. Conversely, CFAV-virus specific reads were identified by first filtering out CFAV-EVEs mapping reads, followed by the alignment of unmapped reads to CFAV virus genome (Bristol strain, accession # KU936054). For the genome-wide analysis of EVE-derived small RNAs, reads were aligned to all EVEs identified in the Aag2 PacBio genome without the CFAV-EVEs to avoid confounding effect from the previous analysis. For the three additional pull-down followed by small RNA-seq experiments, elimination of potential concatenate reads was achieved by only retaining reads that align perfectly (-v 0) to the different reference genomes of interest (e.g.: SINV, TEfam database) over their entire length, starting with a read length of 45 nucleotides. Reads that failed to align were trimmed of 1 base and assessed again for perfect alignment to the reference genome of interest. This process was repeated to the lowest read size of 18 nucleotides.

Reverse transcriptase inhibitor treatments were carried out as described in *Goic et al. (2013)*.

## Transcriptome analysis of Aag2 cells

Aag2 transcripts abundance was calculated using published RNA-seq data from Aag2 cells (*Maringer et al., 2017*) and the Galaxy platform (usegalaxy.org; *Afgan et al., 2016*). RNA-seq reads were aligned and quanti- fied to the Liverpool *Aedes aegypti* transcripts reference (Aedes-aegypti-Liver- pool_TRANSCRIPTS_AaegL3.4.fa, available at www.vector base.org) using the transcript quantification tool, Salmon (*Patro et al., 2017*). Relative transcript abundance is expressed as Transcript Per Million.

Comparative analysis of SINV and Aag2 mRNA-derived sequences in episomal/circular DNA sequencing dataset (see above) was performed using Bowtie 1.2.1.1. To filter out contamination from genomic sequences, episomal DNA reads were first aligned to the new Aag2 genome obtained

**Table 1.** Number of total reads for virus and TE derrived piRNAs.

| Normal | dsCtr | dsAgo2 | dsAgo3 | dsPiwi4 |
|---|---|---|---|---|
| Total reads | 14779383 | 13697808 | 10207667 | 12139804 |
| virus | 194682 | 659432 | 125859 | 301905 |
| TE | 1617084 | 1220933 | 782789 | 1484997 |

DOI: https://doi.org/10.7554/eLife.41244.020

**Table 2.** Number of reads for virus and TE derrived piRNAs following beta elimindation.

| Beta | dsCtr-B | dsAgo2-B | dsAgo3-B | dsPiwi4-B |
|---|---|---|---|---|
| Total reads | 14008617 | 16135731 | 6153485 | 10108849 |
| virus | 242815 | 562820 | 38657 | 23246 |
| TE | 2109794 | 2163585 | 250759 | 331526 |

DOI: https://doi.org/10.7554/eLife.41244.021

by PacBio sequencing (see below), in v mode allowing up to two mismatches (-v 2). SINV and host mRNA-derived sequences were then identified by aligning the genome-unmapped reads to SINV genome (accession # NC_001547) or *Aedes aegypti* transcripts (Aedes-aegypti-Liverpool_TRANSCRIPTS_AaegL3.4.fa) respectively, using the same v mode (-v 2).

## EVE identification

Identification of EVEs was achieved using standalone Blast. Blast Searches were run using the Blastx command specifying the genome as the query and a refseq library composed of the ssRNA and dsRNA viral protein-coding sequences from the NCBI genomes website as the database. The E-value threshold was set at $10^{-6}$.

The EVE with the lower E-value was chosen for further analysis to predict EVEs that overlapped. Several Blast hits to viral protein genes were identified as artifacts because of their homology to eukaryotic genes (e.g. closteroviruses encode an Hsp70 homologue). These artifacts were filtered by hand.

To determine the necessity of genes for the biogenesis of virus RNA-derived piRNAs, we knocked down expression of each gene in Aag2 cells by dsRNA soaking and infected with SINV (MOI = 10). As a control, we soaked Aag2 cells in Fluc dsRNA. Following a four-day infection, we harvested the small RNA (<200 nt) for deep sequencing analysis and large RNA (>200 nt) for RT-qPCR analysis of knockdown efficiency and SINV genome copy number. RT-qPCR analysis of these samples confirmed that knockdown of each gene was effective (>90%) and SINV genome copy number increased with each treatment relative to control.

## PCR and cloning

Genomic DNA was purified from Aag2 cells using Nucleospin Tissue mini spin columns (Macherey-Nagel). Fragments of EVEs were amplified from genomic DNA by PCR using Phusion polymerase (NEB).~500 bp PCR products from EVEs were non-directionally cloned into the 3' UTR of pUb-Renilla using NotI (NEB). Clones of inserts in both sense and antisense polarity were isolated and amplified.

## Data availability

Next-generation sequencing libraries of small RNAs and extrachromosomal circular DNAs are available through NCBI Sequence Read Archive (SRA) at https://www.ncbi.nlm.nih.gov/sra/PRJNA493127.

# Additional information

## Funding

| Funder | Grant reference number | Author |
|---|---|---|
| National Institute of Allergy and Infectious Diseases | R01AI137471 | Patrick T Dolan |
| Defense Advanced Research Projects Agency | Prophecy | Raul Andino |

The funders had no role in study design, data collection and interpretation, or the decision to submit the work for publication.

## Author contributions

Michel Tassetto, Conceptualization, Data curation, Formal analysis, Investigation, Writing—original draft, Writing—review and editing; Mark Kunitomi, Conceptualization, Data curation, Formal analysis, Investigation, Writing—original draft; Zachary J Whitfield, Data curation, Software, Formal analysis, Investigation, Methodology, Writing—review and editing; Patrick T Dolan, Data curation, Software, Formal analysis, Investigation, Writing—original draft, Writing—review and editing; Irma Sánchez-Vargas, Investigation, Methodology; Miguel Garcia-Knight, Investigation, Writing—review and editing; Isabel Ribiero, Taotao Chen, Methodology; Ken E Olson, Data curation, Supervision, Investigation, Writing—review and editing; Raul Andino, Conceptualization, Data curation, Formal analysis, Supervision, Funding acquisition, Writing—review and editing

## Author ORCIDs

Michel Tassetto (iD) https://orcid.org/0000-0001-5666-8796
Mark Kunitomi (iD) https://orcid.org/0000-0002-4626-8972
Zachary J Whitfield (iD) http://orcid.org/0000-0001-9965-428X
Patrick T Dolan (iD) https://orcid.org/0000-0002-4169-0058
Ken E Olson (iD) https://orcid.org/0000-0002-3135-4878
Raul Andino (iD) https://orcid.org/0000-0001-5503-9349

## Decision letter and Author response

Decision letter https://doi.org/10.7554/eLife.41244.025
Author response https://doi.org/10.7554/eLife.41244.026

# Additional files

## Supplementary files

• Transparent reporting form
DOI: https://doi.org/10.7554/eLife.41244.022

## Data availability

Next generation sequencing libraries of small RNAs and extrachromosomal circular DNAs are available through NCBI Sequence Read Archive (SRA) at https://www.ncbi.nlm.nih.gov/sra/PRJNA493127.

The following dataset was generated:

| Author(s) | Year | Dataset title | Dataset URL | Database and Identifier |
|---|---|---|---|---|
| Michel Tassetto, Mark Kunitomi, Raul Andino | 2018 | Sequencing of viral nucleic acids produced during arbovirus infection of Aedes aegypti cells | https://www.ncbi.nlm.nih.gov/sra/PRJNA493127 | NCBI BioProject, PRJNA493127 |

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
