## [Decision Letter]

Thank you for submitting your article "Antiviral adaptive immunity and tolerance in the mosquito *Aedes aegyti*" for consideration by *eLife*. Your article has been reviewed by three peer reviewers, one of whom is a member of our Board of Reviewing Editors, and the evaluation has been overseen by Wenhui Li as the Senior Editor. The reviewers have opted to remain anonymous.

The reviewers have discussed the reviews with one another and the Reviewing Editor has drafted this decision to help you prepare a revised submission. The paper may be further considered for publication in *eLife* only if all the concerns are adequately addressed.

Summary:

This work nicely ties together research from mosquitos showing that piRNAs can function in an antiviral-manner with work showing that extrachromosomal circular DNAs can be produced and retained in hemocytes to confer long-lasting immunity in the fly or mosquito. This work more thoroughly characterizes the role of Piwi4 in the mosquito antiviral immunity, showing that it is associated with antisense, viral piRNAs, and provides the most comprehensive, though not complete, mechanism for their biogenesis described to date. This manuscript also describes a role for Piwi4 to function in somatic tissues to protect the mosquito from persistent infections, which is very distinct from the germline-specific role of Piwi in *Drosophila*.

Essential revisions:

1) This manuscript appears to be sloppily prepared, raising questions about its credibility. It is hard to follow and understand. A few examples throughout the manuscript: The third paragraph of the Introduction ends abruptly; Figure citations at the end of the first paragraph of the subsection “Piwi4 binds to antisense v-piRNAs”, is wrong; The figure legends for Figure 5 do not match with figure; The Figure 7 does not have panel D, which appeared in figure legend. These issues with the figure legends make it impossible to understand some of the data presented without guessing which legends belongs with each figure panel. Relatedly, some of the data presentation is uninterpretable to the generalist reader of *eLife*, especially the presentations in Figure 5 that include the analysis of next-gene sequencing results from circular DNAs. Likewise, the methodology is sometimes very difficult to follow. For example, in Figure 7Ai, how was this expression level of various EVEs determined? Is this PIWI4 pulldown/seq? Or bulk small RNA seq? Or something else? Although the details are in the Materials and methods, the overall approach should be explained in plain English in the main text and ideally be clear from the data presentation in the figure.

2) The statistical analysis is inconsistent throughout. Some figures make claims of significance without providing any statistical metrics, even when effects are relatively small. For example, the modest 3 fold increase in 28bp small RNAs, compared to 22/22bp RNAs is argued to be a significant difference. Likewise, The differences in enrichment between the EVE reads, the TE-assoc. with EVE reads, and the TE-only reads is only about 2x. Statistical demonstration of statistical significance is lacking. Also, the authors just assumed all of their data were not normally-distributed, and performed non-parametric stats without doing normality tests. They should also include statistical test names and test statistic values within the Results section, and real P values, rather than just writing P < 0.01 etc.

3) The paper is often framed from an in vivo perspective. However, ~95% of the data were generated in vitro. Specifically, most data (except Figure 1B and C) were generated based on mosquito Aag2 cells. Since the Aag2 cell line was established in 1968 and was passaged for many generations, whether this anti-viral immunity pathway applies to mosquitoes requires further confirmed. For example, the authors identified a CFAV-derived EVE in Aag2 genome, but it is missing in the Ae. aegypti genome AaegL3. Ideally, data should be provided that takes advantage of a virus that is engineered to contain an EVE element observed in the genome of actual mosquitoes, to perform an in vivo analysis similar to that shown for Aag2 cells in Figure 7B/C.

Likewise, the claims of "transgenerational antiviral immunity" were raised in the Introduction, and given a whole paragraph in the Discussion. Yet, the authors don't have any data supporting this concept. At a minimum, the relevant sections of the paper should be written speculatively.

4) For both the existing cell-based data in Figure 7 (and any additional mosquito based data) the effect of these EVE elements in restricting virus replication needs to be causally linked to Piwi4.

5) How is it determined that the viral gene fragment conversion is uniform (Figure 5D)? Is there a statistical test to show that there is no bias? There appear to be patches with fewer mapped reads (i.e. around 4000 and again around 8000), and areas with more densely mapped reads (1-1000) to my untrained eye. A statistical model showing that these are stochastic is needed to make these claims.

6) Missing and incomplete citations. There are quite a few places in the manuscript lacking references – the Discussion in particular is guilty of this. Other examples include Introduction, third paragraph; subsection “Piwi4 binds to a specific form of v-piRNAs transcribed from vDNA element”, last paragraph; subsection “Ae. Aegypti”.

7) In this paper, the authors propose that piRNAs from virus are produced by some other PIWI proteins, and then get matured in Piwi4 by gaining the 3'-end methylation. This means that the piRNAs need to be transferred from one PIWI protein to Piwi4 for methylation. Since in all examined animals, piRNA production (loading) and methylation are coupled, it is hard for me to imagine how the proposed two step mechanism would occur. Some insight is required.

8) Perhaps related to #7: in the field, it is proposed that Ago3 and Piwi5 are the central players for piRNA biogenesis and viral defense in mosquito somatic cells. Van Rij group actually reported that Piwi4 is not induced upon SINV infection, and Piwi4 does not associate with v-piRNAs (Miesen et al., 2015). In this paper, the authors appear to ignore the potential function of Piwi5 completely, and intentionally only focus on Piwi4. The authors need to have a better explanation on the discrepancy with published work and potentially examine the role Piwi5.

9) For the first half section of the manuscript, the authors focus on SINV and claim that Piwi4 preferentially binds with piRNAs from viruses. But in the second half section, the authors switch to CFAV and propose that Piwi4 mainly binds with piRNA from EVE, but not viruses. The Discussion should address this dichotomy more explicitly; do these two ideas contradict?

Title:

The title is overstated, for the current version of the manuscript, as the antiviral adaptive immunity examined in the manuscript is characterized only in mosquito cell lines.

[Editors' note: further revisions were requested prior to acceptance, as described below.]

Thank you for submitting your article "Control of RNA viruses in mosquito cells through the acquisition of vDNA and endogenous viral elements" for consideration by *eLife*. Your article has been reviewed by two peer reviewers, one of whom is a member of our Board of Reviewing Editors, and the evaluation has been overseen by Wendy Garrett as the Senior Editor. No reviewers found for this submission.

Two reviewers have discussed the reviews with one another and the Reviewing Editor has drafted this decision to help you prepare a revised submission. The third reviewer was unavailable.

Summary:

This work nicely ties together research from mosquitos showing that piRNAs can function in an antiviral-manner with work showing that extrachromosomal circular DNAs can be produced and retained in hemocytes to confer long-lasting immunity in the fly or mosquito. This work more thoroughly characterizes the role of Piwi4 in the mosquito antiviral immunity, showing that it is associated with antisense, viral piRNAs, and provides the most comprehensive, though not complete, mechanism for their biogenesis described to date. This manuscript also describes a role for Piwi4 to function in somatic tissues to protect the mosquito from persistent infections, which is very distinct from the germline-specific role of Piwi in *Drosophila*. The revised manuscript has greatly improved the data presentation issues, highlighted in the earlier round of reviews and the new title more accurately reflects the conclusions.

Essential revisions:

One major issue remains, that was brought up in the earlier reviews, and still requires attention. In particular, a key prediction of the models is that control of the engineered SINV, containing the sense-strand of the CFAV EVE, should be dependent on Piwi4. This needs to be tested. It should be a straight forward experiment given the tools already in-hand.

---

## [Author Response]

Essential revisions:1) This manuscript appears to be sloppily prepared, raising questions about its credibility. It is hard to follow and understand. A few examples throughout the manuscript: The third paragraph of the Introduction ends abruptly; Figure citations at the end of the first paragraph of the subsection “Piwi4 binds to antisense v-piRNAs”, is wrong; The figure legends for Figure 5 do not match with figure; The Figure 7 does not have panel D, which appeared in figure legend. These issues with the figure legends make it impossible to understand some of the data presented without guessing which legends belongs with each figure panel. Relatedly, some of the data presentation is uninterpretable to the generalist reader of eLife, especially the presentations in Figure 5 that include the analysis of next-gene sequencing results from circular DNAs. Likewise, the methodology is sometimes very difficult to follow. For example, in Figure 7Ai, how was this expression level of various EVEs determined? Is this PIWI4 pulldown/seq? Or bulk small RNA seq? Or something else? Although the details are in the Materials and methods, the overall approach should be explained in plain English in the main text and ideally be clear from the data presentation in the figure.

In this revised manuscript we have paid particular attention to those issues and details, and we hope that we have identified and corrected them. For example, we have revised all the references to the figures and other typos as well as detailed some parts of the Materials and methods. To better explain the next generation sequencing of circular DNAs, we have added detailed cartoons in Figure 5 describing our approaches to highlight what was sequenced. Regarding the expression levels of piRNAs from various EVEs, the legend of Figure 7A has been edited to explain that EVE-derived piRNAs are expressed as a ratio to total read number (i.e. bulk small RNA seq). Importantly, all the results presented are consistent with a model where Piwi4 is a central player in the pathway, whereas Piwi5/Ago3 are accessory proteins that generate sense small viral RNAs, but that do not provide antiviral protection.

2) The statistical analysis is inconsistent throughout. Some figures make claims of significance without providing any statistical metrics, even when effects are relatively small. For example, the modest 3 fold increase in 28bp small RNAs, compared to 22/22bp RNAs is argued to be a significant difference. Likewise, The differences in enrichment between the EVE reads, the TE-assoc. with EVE reads, and the TE-only reads is only about 2x. Statistical demonstration of statistical significance is lacking. Also, the authors just assumed all of their data were not normally-distributed, and performed non-parametric stats without doing normality tests. They should also include statistical test names and test statistic values within the Results section, and real P values, rather than just writing P < 0.01 etc.

IP enrichment analysis now includes a bootstrapping of each dataset (resampling of 10E6 reads repeated 10E4). From this, we inferred the corresponding 95% confidence intervals and found that our experimental data are within the confidence interval. Based on these confidence intervals, we showed that there is a 95% confidence level that v-siRNAs (20-22nt) are less enriched than piRNAs (27-28nt) in Piwi4 pull down experiments.

For all other experiments (that is, not Next-Generation Sequencing datasets), the sample size was n=4. Normal distribution was assessed with the Shapiro-Wilk normality test in R (shapiro.test()). No sample was found to have normal distribution.

Therefore, all the comparisons of mean values were analyzed by Wilcox ran sum test, a non-parametric test fitting this sample size. Because all sample sizes were equal (n=4), the Wilcox rank sum tests gave the same p value of 0.02857 (p<0.05) for each test.

3) The paper is often framed from an in vivo perspective. However, ~95% of the data were generated in vitro. Specifically, most data (except Figure 1B and C) were generated based on mosquito Aag2 cells. Since the Aag2 cell line was established in 1968 and was passaged for many generations, whether this anti-viral immunity pathway applies to mosquitoes requires further confirmed. For example, the authors identified a CFAV-derived EVE in Aag2 genome, but it is missing in the Ae. aegypti genome AaegL3. Ideally, data should be provided that takes advantage of a virus that is engineered to contain an EVE element observed in the genome of actual mosquitoes, to perform an in vivo analysis similar to that shown for Aag2 cells in Figure 7B/C.Likewise, the claims of "transgenerational antiviral immunity" were raised in the introduction, and given a whole paragraph in the Discussion. Yet, the authors don't have any data supporting this concept. At a minimum, the relevant sections of the paper should be written speculatively.

We agree with the reviewer that further experimentation will be required to establish the precise EVEs role in antiviral defense in vivo. We have modified accordingly the relevant sections to more clearly specify that our data show that EVEs produce functional antiviral piRNAs in a Piwi4-dependent manner, and we suggest that given that Aag2 cells are persistently infected with CFAV, EVEs are under positive selection and are maintained during cell divisions in the mosquito cell line. This leads to the speculation that similar positive selection could exist in mosquito populations. Indeed, previous reports showing the presence of different EVEs in distinct mosquito species collected from the same geographic area suggests that EVEs are endogenized and maintained in mosquito populations in part in a lineage-specific manner (Palatini et al., 2017). We further speculate that EVEs might originate from ancient vertically transmitted insect-specific viruses. However, even though transgenerational EVE-mediated immunity could be a low-frequency event, during acute arboviral infection the acquisition of EVEs in somatic cells, such as mosquito circulating immune cells (i.e.: hemocytes), may be central to effective antiviral immunity. Thus, our results presented in this study represent an important advance in our understanding of the mechanism of mosquito piRNA immunity.

4) For both the existing cell-based data in Figure 7 (and any additional mosquito based data) the effect of these EVE elements in restricting virus replication needs to be causally linked to Piwi4.

CFAV data address this issue. Piwi4 binds only to CFAV-EVE derived piRNAs (Figure 6D and E) and knock-down of Piwi4, which leads to a decrease of CFAV-EVE derived piRNAs, significantly increases CFAV genomic RNA (Figure 7C). Thus, EVE-derived piRNAs mediate the antiviral effect and Piwi4 is a central player in the pathway.

5) How is it determined that the viral gene fragment conversion is uniform (Figure 5D)? Is there a statistical test to show that there is no bias? There appear to be patches with fewer mapped reads (i.e. around 4000 and again around 8000), and areas with more densely mapped reads (1-1000) to my untrained eye. A statistical model showing that these are stochastic is needed to make these claims.

Indeed, the distribution of SINV viral gene fragments is not uniform over its genome. This observation was confirmed using Kolmogorov-Smirnov statistic to test for uniform distribution of reads, which indicated that the distribution of vDNA reads is not uniform across the genome. We have corrected the manuscript to better describe the data, namely by removing the mention of “uniform distribution” and instead indicates that SINV viral RNA genome is almost fully converted into DNA, with only a couple of specific genomic regions presenting lower density of vDNA fragments libraries. This might be due to structure and/or initiation site in the subgenomic viral RNA promoter, and depth of sequencing.

6) Missing and incomplete citations. There are quite a few places in the manuscript lacking references – the Discussion in particular is guilty of this. Other examples include Introduction, third paragraph; subsection “Piwi4 binds to a specific form of v-piRNAs transcribed from vDNA element”, last paragraph; subsection “Ae. Aegypti”.

In the revised manuscript we have made an effort to correct the deficiencies pointed out by the reviewer and to include as many relevant references as possible.

7) In this paper, the authors propose that piRNAs from virus are produced by some other PIWI proteins, and then get matured in Piwi4 by gaining the 3'-end methylation. This means that the piRNAs need to be transferred from one PIWI protein to Piwi4 for methylation. Since in all examined animals, piRNA production (loading) and methylation are coupled, it is hard for me to imagine how the proposed two step mechanism would occur. Some insight is required.

We hypothesize that the production of the Piwi4 bound v-piRNAs is akin to primary anti-transposon piRNA production, where piRNA precursors are transcribed from piRNA clusters and processed by an endonuclease (e.g.: Zucchini in *Drosophila*) to produce primary guide piRNAs that are loaded onto Piwi where they become methylated. In our model, v-piRNA precursor are first transcribed from vDNA molecules (either as episomal and/or chromosomal EVEs), these vDNA-derived piRNAs are then processed by a yet unknown endonuclease before being loaded onto Piwi4, followed by methylation. We have now added a section in our Discussion to further clarify these aspects of the model.

8) Perhaps related to #7: in the field, it is proposed that Ago3 and Piwi5 are the central players for piRNA biogenesis and viral defense in mosquito somatic cells. Van Rij group actually reported that Piwi4 is not induced upon SINV infection, and Piwi4 does not associate with v-piRNAs (Miesen et al., 2015). In this paper, the authors appear to ignore the potential function of Piwi5 completely, and intentionally only focus on Piwi4. The authors need to have a better explanation on the discrepancy with published work and potentially examine the role Piwi5.

The antiviral role of Piwi5 was tested in our study and we found that Piwi5 doesn’t have an antiviral function under the experimental condition assayed (Figure 1—figure supplement 1A). This is consistent with previously reported results by the Van Rij group. To clarify this issue, we now included a discussion and modified the model in Figure 7 on the potential role of Piwi5 in the process. Briefly, we propose Piwi5 is involved in the generation of small RNAs from degraded viral genomic RNA, whereas Piwi4 is involved in vDNA-derived functional piRNAs biogenesis. Since Dicer2, Ago2 and Piwi4 act as antiviral factors but not Piwi5 in our screen (and in a previous study on Piwi5 (Miesen et al., 2015)), we hypothesize that the Dicer2/Ago2 pathway directly targets viral RNA genomes, which in turns triggers vDNA production, similar to the antiviral immune response in *Drosophila* (Tassetto et al., 2017 and Poirier et al., Cell Host and Microbe, 2018). piRNAs transcribed from vDNA are then loaded on Piwi4, which participate in the control of virus replication. Given that Piwi5 and Ago3 are involved in the production of virus-derived piRNA production (Miesen el al., 2015), but lack antiviral function, we suggest that these factors participate in degradation of viral RNA genomes fragments that have been already targeted and inactivated by Ago2 and Piwi4 (Figure 7D).

9) For the first half section of the manuscript, the authors focus on SINV and claim that Piwi4 preferentially binds with piRNAs from viruses. But in the second half section, the authors switch to CFAV and propose that Piwi4 mainly binds with piRNA from EVE, but not viruses. The Discussion should address this dichotomy more explicitly; do these two ideas contradict?

The first half of our manuscript describes the role of Piwi4 in acute arboviral infections, especially in the case of SINV infection. We show that Piwi4 binds to antisense SINV-derived piRNAs and is required for their methylation. We also demonstrate that Piwi4 acts as antiviral factor against SINV (and also CHIKV and DENV2) in cell culture and in adult female mosquitoes (in the case of DENV2). We then demonstrate that inhibition of vDNA production in mosquito cells leads to a decrease in v-piRNA production and an increase in viral replication. This suggests that the antiviral piRNAs bound by Piwi4 originate from vDNA transcription. During acute viral infection, the reverse-transcription of viral RNA genomes into vDNA leads to the production of two viral nucleic acid elements indistinguishable by deep-sequencing. In contrast, by re-sequencing the genome of Aag2 cells, we identified endogenized viral elements (EVEs) from a Cell Fusing Agent Virus (CFAV) strain distinct from the related CFAV strain that persistently infects Aag2 cells. We thus leveraged our ability to distinguish between piRNAs derived from CFAV EVEs and those derived from replicating CFAV to further our understanding of Piwi4 antiviral role. We found that Piwi4 binds specifically to piRNAs derived from EVEs and not from the replicating RNA virus. Since Piwi4 restricts viral replication during acute and persistent infection, and vDNA formation is involved in antiviral defense, our results suggest that Piwi4 binds to vDNA-derived piRNAs in both acute and persistent infections.

Title:The title is overstated, for the current version of the manuscript, as the antiviral adaptive immunity examined in the manuscript is characterized only in mosquito cell lines.

Title has been changed to:

Control of RNA viruses in mosquito cells through the acquisition of viral DNA and endogenous viral elements

[Editors' note: further revisions were requested prior to acceptance, as described below.]Essential revisions:One major issue remains, that was brought up in the earlier reviews, and still requires attention. In particular, a key prediction of the models is that control of the engineered SINV, containing the sense-strand of the CFAV EVE, should be dependent on Piwi4. This needs to be tested. It should be a straight forward experiment given the tools already in-hand.

To address the essential revision identified by the Editors on the role of Piwi4 in EVE-piRNA antiviral activity, we have now conducted an experiment in 4 independent biological replicates using Sindbis virus strains engineered to express the *firefly luciferase* reporter gene and a sense or antisense EVE target sequence, in the context of Piwi4 knock down in Aag2 cells. Specific inhibition of the sense EVE target containing virus was assessed as the ratio of luciferase activity from the virus strain with the antisense EVE (AS) over the strain with the sense EVE (S). In control cells (treated with dsRNA against eGFP), the relative viral replication advantage of the AS EVE containing virus over its S EVE counterparts increased over a 4 day time course. On the contrary, this ratio was significantly reduced at 48, 72 and 96 hours post infection (hpi) in Piwi4 knock-down Aag2 cells (Wilcoxon Signed Rank test, n=4 independent experiments). These results are now in Figure 7—figure supplement 2, and confirm that Piwi4 is involved in the specific inhibition of viruses containing EVE-piRNA targets.

Taken together, our results indicate that EVE-piRNAs are loaded onto Piwi4 and can exert sequence-specific antiviral immunity in Aedes aegypti cells.

In the absence of any available specific Piwi4 antibody, we used overexpression of a tagged Piwi4 construct to perform Piwi4 pull down followed by small RNA-seq to identify small RNA species associated with Piwi4. If overexpression was to increase non-specific binding of piRNAs by the tagged Piwi4, one would expect to see a bias for the most abundant small RNA species, in origin and/or size. However, we have provided 3 levels of control that indicate that our data are very likely representative of the behavior of the endogenous Piwi4.

First, we found that Piwi4 preferentially binds to EVE-derived and virus-derived piRNAs and not to TE-piRNAs, whereas the latter are the most abundant piRNA species present in Aag2 cells.

Second, we found that Piwi4 preferentially binds to longer form of virus-derived small RNAs (piRNA vs. siRNA), whereas the virus-derived siRNAs are more abundant than virus-derived piRNAs.

Last, we confirmed the absence of bias in our approach by normalizing to total small RNA input, performing three independent pull down experiments with the tagged Piwi4 and one control experiment with eGFP. Ultimately, we performed boostrapping of all datasets to confirm that the size distribution of small RNAs bound to Piwi4 was specific and not biased by the amount of small RNA isolated by tagged Piwi4 pull down (with a 95% confidence interval).

In conclusion, we are confident that the description of our experimental approach and statistical analysis in the Results and Materials and methods sections address the concerns about potential bias due to tagged Piwi4 overexpression at the relevant points in our study and would not improve the Discussion section.